



# Contrasting physical properties of black carbon in urban Beijing between winter and summer

Dantong Liu[1,2], Rutambhara Joshi[2,5], Junfeng Wang[4], Chenjie Yu[2], James D. Allan[2,5], Hugh Coe[2], Michael J. Flynn[2], Conghui Xie[3], James Lee[6], Freya Squires[6], Simone Kotthaus[7], Sue Grimmond[7], Xinlei Ge[4], Yele Sun[3], Pingqing Fu[3]

[1] Department of Atmospheric Sciences, School of Earth Sciences, Zhejiang University, Hangzhou, Zhejiang, China

[2] Centre for Atmospheric Sciences, School of Earth and Environmental Sciences, University of Manchester, Manchester, UK

[3] Institute of Atmospheric Physics, Chinese Academy of Sciences, Beijing, China

[4] School of Environmental Science and Engineering, Nanjing University of Information Science and Technology, Nanjing, China

[5] National Centre for Atmospheric Science, University of Manchester, Manchester, UK

[6] Department of Chemistry & National Centre for Atmospheric Science, University of York, York, UK

[7] Department of Meteorology, University of Reading, UK

Corresponding to: Dantong Liu (dantongliu@zju.edu.cn)



## Abstract

Black carbon (BC) is known to have major impacts on both human health and climate. The populated megacity represents the most complex anthropogenic BC emissions where the sources and related impacts are very uncertain. This study provides source attribution and characterization of BC in the Beijing urban environment during the joint UK-China APHH (Air Pollution and Human Health) project, in both winter (Nov. - Dec. 2016) and summer (May - Jun. 2017). The size-resolved mixing state of BC-containing particles was characterized by a single particle soot

photometer (SP2) and their mass spectra was measured by a soot particle mass spectrometer (SP-AMS). The refractory BC (rBC) mass loading was around a factor of 2 higher in winter relative to summer and more variable coatings were present, likely as a result of additional surface emissions from the residential sector and favourable condensation in cold season. The characteristics of the BC were relatively independent of air mass direction in summer; whereas in winter the airmass from the Northern Plateau had a significant dilution effect resulting in less-

coated and smaller BC, whereas the BC from the Southern Plateau had the largest core size and coatings.

We combine two online source apportionment methods for the first time, by the physical method from the SP2, and the chemical approach using the positive matrix factorization (PMF) of mass spectra from the SP-AMS. A method is proposed to isolate the BC from the transportation sector using a mode of small BC particles (core diameter $D_c$<0.18µm and coating thickness $ct$ <50nm). This mode of BC highly correlated with $NO_x$ concentration in both

seasons (~14 ng m$^{-3}$ BC ppb$^{-1}$ $NO_x$) and corresponded with the morning traffic rush hour, contributing about 30% and 40% of the total rBC mass (35% and 55% in number) in winter and summer respectively. The BC from coal burning or biomass burning tended to dominate with moderate coatings ($ct$=50-200nm) contributing ~20-25% of rBC mass. Large uncoated BC particles ($D_c$>0.18µm and $ct$<50nm) was more likely to be contributed by coal combustion, as these particles were not present in urban London. This mode was present in Beijing in both winter (~30-40% rBC

mass) and summer (~40% rBC mass) but may be dominated by residential and industrial sector respectively. The contribution of BC thickly-coated with secondary species ($ct$>200nm) to the total rBC mass increased with pollution level in winter, but was minor in summer. These large BC importantly enhanced the absorption efficiency at high pollution levels - in winter when PM1>100 µg m$^{-3}$ or BC>2 µg m$^{-3}$, the absorption efficiency of BC increased by 25-70%. Reduction of emissions of these large BC particles and the precursors of the associated secondary coating will

be an effective way of mitigating the heating effect of BC in urban environments.



# 1 Introduction

Black carbon aerosol (BC) has a significant impact on both climate (Bond et al., 2013) and human health
(Baumgartner et al., 2014). Its regional impact in the atmosphere may be very large, especially close to polluted
hotspots such as in South and East Asia, where anthropogenic emissions are high and population exposure is severe
(Ramanathan and Carmichael, 2008). It has been estimated that BC over China could contribute up to 14% of the
global radiative forcing budget (Li et al., 2016). Reducing BC has been postulated as a win-win policy intervention
because of the shorter atmospheric lifetime of BC compared to the greenhouse gases, delivering immediate mitigation,
while at the same time improving air quality (Kopp and Mauzerall, 2010).

Beijing, as one of the most populated megacities in the world has experienced severe air pollution (Yang et al.,
2005;Xu et al., 1994). The complexity of emissions from multiple sectors that are often co-located (Li et al., 2017)
make it extremely challenging to attribute source contributions to the BC load, hindering policy making on emission
regulations. The source apportionment of BC in urban environments has been studied using both online and offline
measurements and using site receptor models (Cao et al., 2005;Viana et al., 2008 and refs therein). Most of the
techniques separate the fossil fuel BC, such as that from traffic sources from the solid fuel burning fraction (such as
that from wood burning (Sandradewi et al., 2008;Healy et al., 2012) or open biomass burning (Schwarz et al., 2008).
These techniques include using biomass burning tracers in aerosol (Puxbaum et al., 2007), or using individual organic
tracer compounds to attribute the sources and provide time series representing the different sources. These are then
used to segregate the different BC emission contributions by multi-linear regression (Liu et al., 2011;Laborde et al.,
2013). An approach using Aethalometer measurements has been widely used based on the spectral dependence of
absorption (Sandradewi et al., 2008). This technique needs to assume a prescribed absorption spectrum from traffic
or wood burning sources which may be subject to variation under different burning conditions (Zotter et al., 2017).
Isotope analysis of elemental carbon, in conjunction with thermal separation, allows identification of modern carbon
(e.g. from biomass) from fossil fuel (from diesel or coal) based on the $^{14}$C abundance (Bernardoni et al., 2013;Zhang
et al., 2012). This method has been considered to be relatively unambiguous in isolating wood burning sources from
traffic source, and has been used to validate the other methods in attributing elemental carbon (Liu et al., 2013;Zotter
et al., 2017).

These techniques, mostly use distinct features in the chemistry or physical properties to isolate one BC source from
the other, and to do so requires that there are unique characteristics that are separable. The BC sources in Beijing are
combinations of residential, industrial and transport sectors (Li et al., 2017), and the fuel use could be rather more
complex than two distinct sources which most techniques are based on. For example, both coal burning and diesel
fuel could emit fossil fuel BC, which may not be isolated through isotope analysis, and also the absorption spectrum
of BC from different sources may vary considerably and to assume a single pattern based on Aethalometer
measurements may not be suitable to attribute multiple sources. The fast secondary processing of aerosols in Beijing
(Sun et al., 2016b) may make the source attribution of primary BC even more challenging as the secondary coating
formed on BC may alter its original source-dependent features. Given these difficulties it is unlikely that any single



methodology will give unambiguous results, but a combination of different methods may improve the understanding on the source attribution because source-specific physio-chemical properties of BC may be reflected in different ways by different methods.

It is also necessary to gain knowledge on the microphysical structure and mixing state of the soot, namely its size and what other materials are present on the individual particles, as these dictate its impact on the wider atmosphere. Other material present on a BC particle (a 'coating') may alter its optical properties (Liu et al., 2017), affecting the direct radiative effect on the local atmosphere, and it may also make it more susceptible to in-cloud scavenging, meaning that it can perturb the cloud properties or experience a shortened atmospheric lifetime through wet deposition (Hodnebrog et al., 2014). The source profile of size-resolved mixing state of BC is desired for the evaluation of BC properties in process models (Riemer et al., 2009) especially for environments with combined sources (Fierce et al., 2016).

This study quantifies the source attribution of BC-containing particles in urban Beijing and delivers source-specific information on their properties by combining two novel techniques, both of which directly characterize BC-containing particles but are based on physical and chemical techniques respectively. The physical technique uses a single particle soot photometer (DMT, SP2), which was previously used for source apportionment of BC in urban London (Liu et al., 2014b). This approach is to examine mixing state of BC particles as a function of their core size and this has been used to attribute the BC from traffic diesel and wood burning source. The chemical approach uses the soot particle aerosol mass spectrometer (Aerodyne, SP-AMS) which has been previously used to identify the chemical compositions of coatings associated with BC (Onasch et al., 2012a), which may be used to determine primary sources or secondary processing of BC. The combination of both techniques in this study will give the detailed physio-chemical properties of BC influenced by mixed sources. In particular, by combining the approaches the contribution of different emission sources to the optical properties of BC can be quantified and so an attribution of different sources to the BC heating in the atmospheric column can be made.

## 2 The site, meteorology and air mass classification

The experiments were conducted at the tower site of Institute of Atmospheric Physics (IAP), Chinese Academy of Sciences (39°58′28″N, 116°22′16″E) in Beijing during both winter (Nov.-Dec. 2016) and summer (May-Jun. 2017) periods, as part of the Air Pollution and Human Health-Beijing campaign. This site represents the typical urban Beijing environment with pollution influences from surrounding traffic, commercial activities, residential activities such as cooking and home heating and regional transport (Sun et al., 2016a).

Fig. 1a shows the terrain of the North China Plain (NCP) region to the north of Beijing and Fig. 1b shows the surface emission inventory of BC for the year 2010 (Li et al., 2017). The high anthropogenic BC emission can be generally divided by the border along Taihang and Yanshan Mountain Ridges, beyond which the region from northwest Beijing has relatively lower emissions. Considerably higher emissions are also present across the southern Plateau region. In



order to investigate the regional influence of pollutants in Beijing, the regions over the NCP and the Plateau are classified according to the terrain and BC surface emission, shown in Fig. 1c. The regions are firstly classified as the plateau and plain according to the terrain height below and above 800m, then for the region <800m, 116.5°E (the longitude of central Beijing) is used to separate the Eastern and Western NCP (E and W are used as the abbreviations during the following discussion); the region >800m is separated as Northern and Southern Plateau (N and S are used as abbreviations for the following discussion) using the border along 41.5°N; the Ocean is defined as the terrain height below zero. The Northern Plateau has significantly lower emission, meaning north-westerly air mass will bring clean air into Beijing and are likely to reduce pollution levels in the city, whereas westerlies may bring pollutants from the Western Plateau and south-westerlies could transport the high emissions from central China to the NCP. The local area is defined in this work as the area within the square ±0.25° away from the measurement site. BC emission inventories from different sectors are shown in Fig. S1. In winter the residential sector, which is mainly composed of residential coal burning, contributes the vast majority of BC emissions; whereas the emissions from the industrial (which also contains significant coal consumption) and transportation sectors are maintained throughout the year. This means the differences in BC emissions between winter and summer will mainly result from changes in the residential sector.

The HYSPLIT backtrajectory model (Draxler and Hess, 1998) was run using the 1°×1° horizontal and vertical wind fields provided by the GDAS1 reanalysis meteorology. Given the emissions are intensive around the immediate Beijing area, air mass back trajectories were only followed for the previous 24-hours to examine the influence from the most recent air masses. The back trajectories are then mapped onto the classified regions (Fig. 1c) to investigate variations in the potential regional source influence. The back trajectories have a 1h time resolution and each point along a single 24 hour trajectory is assigned to one of the four regional classifications. All the points along a single trajectory are then used to determine the fraction of time during the previous day that the air mass spent above each of the classified regions as shown in the bottom panel of Fig. 3a and Fig. 3b. This method has been applied previously in the western Africa region to identify the potential source contributions (Liu et al., 2018). Note that the HYSPLIT analysis is not able to reproduce the dispersion of the air mass, but such effects are likely to be minimal since the trajectories are only investigated over the previous 24 hours.

Each back trajectory is assigned to be predominately from one of the regions based on a ranked classification scheme that takes account of the likely greater influence on the pollution at the receptor from closer regions with large emissions. The methodology considers each region in turn, beginning with the western NCP air mass because it represents the mostly polluted region and a relatively lower air mass fraction of western NCP will make an important contribution to the pollutants measured at the receptor. If the back trajectory spent more than 10% of the previous 24 hours over the western NCP it was classified as being from that sector. The following regions are then considered in a similar way in turn based on the following order: eastern NCP, southern Plateau and northern Plateau with each above 10% air mass fraction. Lastly if the air mass spent more than 20% in the local area it is defined as having significant local influence. The fraction of 10% was chosen because by varying this threshold air mass fraction ±10%, we found this is the optimum metric to reflect the air mass influence from the classified regions.



The meteorological parameters such as wind, relative humidity (RH) and temperature were measured at ground level (z=10m) and also on the tower at z=120m. The temperature and RH at z=10m is used, but the wind at 120m is used

to avoid the surface friction effect. In addition, the mixing layer height (MLH) spanning the experimental period was determined using Lidar and Ceilometer measurements (Kotthaus and Grimmond, 2018).

## 3 Instrumentation and data analysis

### 3.1 The physical properties of BC

The physical properties of individual refractory BC particles were characterized using a single particle soot photometer (SP2) manufactured by DMT Inc (Boulder, CO, USA). The instrument operation and data interpretation procedures are described elsewhere (Liu et al., 2010;McMeeking et al., 2010). The SP2 incandescence signal was calibrated for rBC mass using Aquadag® black carbon particle standards (Aqueous Deflocculated Acheson Graphite, manufactured by Acheson Inc., USA) and corrected for ambient rBC with a factor of 0.75 (Laborde et al., 2012). The

mass-equivalent diameter of the rBC core ($D_c$) is obtained from the measured rBC mass assuming a density of 1.8 g m$^{-3}$ (Bond and Bergstrom, 2006). For a given time window, the mass median dimeter (MMD) of rBC core is calculated from the $D_c$ distribution below and above which the rBC mass was equal.

The scattering signal of each BC particle measured by the SP2 is determined using a leading edge only (LEO) technique to reconstruct the distorted scattering signal when the particle passes through the SP2 laser beam (Gao et

al., 2007). This was used to determine the scattering enhancement ($E_{sca}$) for each single particle which is defined as the ratio between the measured scattering of the BC particle, including any coating, and the calculated scattering resulting from the uncoated BC core (Liu et al., 2014a;Taylor et al., 2015), expressed as:

$$E_{sca} = \frac{S_{measured,coated\,BC}}{S_{calculated,uncoated\,BC}} \qquad (1),$$

where the numerator is the scattering of coated BC directly measured by the SP2 and the denominator is the calculated

scattering of uncoated BC core using a refractive index of BC 2.26+1.26i at the SP2 laser wavelength, λ=1064nm (Moteki et al., 2010). For a given $D_c$, a higher $E_{sca}$ means a thicker coating and $E_{sca}$=1 means there is no coating. The coated BC particle size ($D_p$) is then determined by matching the modelled scattering with the measured scattering by applying a Mie core-shell lookup table (Taylor et al., 2015).

The bulk relative coating thickness ($D_p/D_c$) in a given time window is calculated as the total volume of coated BC

particles divided by the total volume of the rBC cores (Liu et al., 2014a), expressed as:

$$\frac{D_p}{D_c} = \sqrt[3]{\frac{\sum_i D_{p,i}^3}{\sum_i D_{c,i}^3}} \qquad (2),$$



where $D_{p,i}$ and $D_{c,i}$ are the coated and rBC diameters for each single particle respectively. Note that the bulk $D_p/D_c$ is largely independent of the uncertainties arising from smaller particles because of their less important contribution to the integrated volume.

The volume-weighted coated BC size ($D_{p,v}$) is then calculated as the product of the bulk relative coating thickness and the MMD of the BC cores, to indicate the mean coated BC size (Equation 3). The bulk mixing ratio of coating mass over rBC mass ($M_{R,bulk}$) can be also derived from $D_p/D_c$ assuming a density of coating and rBC (Equation 4):

$$D_{p,v} = \frac{D_p}{D_c} \times MMD \qquad (3),$$

$$M_{R,bulk} = \left( \left(\frac{D_p}{D_c}\right)^3 - 1 \right) \times \frac{\rho_{coating}}{\rho_{rBC}} \qquad (4),$$

The mass absorption cross section at λ=550nm ($MAC^{550}$) is calculated for each single particle by assuming the refractive index of rBC core 1.95+0.79i (Bond and Bergstrom, 2006) and coating refractive index 1.50+0i (Liu et al., 2015), using the Mie core-shell approach (Bohren and Huffman, 2008). Note that the absorption enhancement due to coating is considered to only occur when the coating mass over rBC mass is larger than 3 according to the recent study of (Liu et al., 2017). Fig. S3 gives the calculated $MAC^{550}$ mapped on the $E_{sca}$-$D_c$ plot. The $MAC^{550}$ in bulk for

a given time window is calculated as the integrated absorption coefficient ($MAC \times m_{rBC}$) for all particles divided by the integrated particle masses, expressed by Equation (5),

$$MAC = \frac{\sum_i MAC_i \times m_{rBC,i}}{\sum_i m_{rBC,i}} \qquad (5),$$

where $MAC_i$ and $m_{rBC,i}$ are the MAC and rBC mass for each single particle respectively. This calculation is performed for each type of BC.

A PAX (Droplet Measurement Technologies, Boulder, CO, USA) (Wang et al., 2014;Selimovic et al., 2018) was deployed to directly measure the in-situ aerosol light absorption every minute using photoacoustic technology. The light-absorbing particles are heated by a laser in the acoustic chamber, and this heating produces pressure waves which are detected by a microphone. The absorption coefficient at λ=870nm ($\sigma_{abs,870}$) is measured by the PAX. The mass absorption cross section (MAC) is determined as the absorption coefficient per unit rBC mass. Note that in this

study $\sigma_{abs,870}$ values for rBC mass loadings <0.5 µg m$^{-3}$ were not used for MAC calculations due to the large uncertainty of absorption measurement at low concentration.

## 3.2 BC chemical composition

The chemical composition of black carbon containing particles, including the refractory BC and coating compositions, are measured by a soot particle mass spectrometer (SP-AMS) (Onasch et al., 2012a;Wang et al., 2017). The results






from SP-AMS measurement during APHH are detailed in Wang et al., 2018. The SP-AMS was run in laser-only mode and so only detected compositions for BC-containing particles. In this mode the non-refractory components were not detected if they are not contained within a BC particle. The ionization efficiency (IE) and relative ionization efficiency (RIE) of sulphate and nitrate were calibrated by using ammonium nitrate and ammonium sulphate (Jayne

et al., 2000). RIE of rBC was calibrated by using Regal Black (RB, REGAL 400R pigment black, Cabot Corp.) (Onasch et al., 2012b). Positive matrix factorization (PMF) (Paatero and Tapper, 1994) was applied to the mass spectra of the organic and rBC components to attribute the source contribution of BC-containing particle mass in real time, as detailed in Wang et al., 2018. Four types of BC-containing particle associated with different organic coatings were identified: fossil fuel combustion OA (FOA), biomass burning OA (BBOA), less-volatile organics (OOA1) and

semi-volatile organics (OOA2) were identified. In addition, a PAH factor was also derived from the SP-AMS measurement, which is associated with coal combustion (Sun et al., 2016a).

## 4 Results

### 4.1 Overview of BC physical properties

Fig. 3 shows the temporal variation of the physical properties of BC, associated gaseous pollutants and meteorological parameters and their association with the air mass classifications. As the bottom panels show, the site was mostly influenced by northerly air masses in winter, and very few air masses came from the eastern NCP. The first half of the winter campaign up to 20[th] Nov. was periodically influenced by air masses from western NCP, and during the second half the synoptic meteorology shifted appreciably and was dominated by northerly (from the northern Plateau)

or westerly (from the southern Plateau) air masses, with the period between 02/12 and 04/12 dominated by air from the southern Plateau. The temperature dropped from ~10ºC to below 5ºC when the air mass type shifted to deliver air from the plateau. In summer, Beijing received air from the western NCP, the eastern NCP and the northern plateau with southerly air masses more dominant than those from the north.

The classified air mass types are generally consistent with the local wind directions measured at 120m for both

seasons (Fig. 4). In winter, the northern Plateau air masses were characterised by high speed, dry NW winds, in summer the flow was not as strong. Air masses from the southern plateau were associated with both northerly and southerly winds but with much lower wind speed and systematically higher RH than the northern plateau air masses. The south-westerly air masses also had the highest RH, which is consistent with previous observations that show air masses from lower latitudes contained more moisture in wintertime (Tao et al., 2012). In summer, air masses from

the western NCP showed lower RH which may result from the almost latitudinally homogenous distribution of higher temperatures. The site showed lower wind speed and wider variation of RH when influenced by local air masses.

BC properties associated with different air mass types in both the winter and summer seasons are compared in Fig. 5. rBC mass loadings were higher in winter than in summer by around a factor of 2 for both local and regionally



transported air masses due to higher surface emissions from both the local Beijing region and the surrounding area in the cold season. However, air masses from the northern Plateau during periods of strong and dry wind had notable effect on the rBC mass in winter, greatly reducing the rBC mass in Beijing. This air mass type contributed to over 90% of the cleaner days (rBC mass concentration<1µg m$^{-3}$) in winter. This is consistent with the emission inventory that the northwest Beijing is dominated by lower surface emissions over Mongolia. The dilution of Beijing pollution by high wind speeds during NW air flow has been widely observed (Sun et al., 2015;Zhang et al., 2013;Zhang et al.,

2015). The BC particles during these periods had systematically lower core sizes (Fig. 5b1) and lower coatings (Fig. 5c1) and therefore smaller total particle sizes. This may be also as a result of more favourable removal processes for the more coated and larger BC particles in winter.

No significant differences in the physical properties of BC particles were observed between the different air mass types in summer, e.g. there is a consistent peak of $D_p/D_c$~1.4 for all air mass types, which may suggest an almost

homogenous mixture or consistent BC sources across a large region around Beijing in China. In winter, the range of $D_p/D_c$ values extended from similar values to those in the summer up to 2.5. The MMD of BC cores was most often observed to be ~180nm for both seasons, but it is noted that the air mass from the southern Plateau (as the green lines show) had systematically larger MMD, and BC particles in air masses from this region also had the highest coatings and largest coated BC size compared to other air masses. The observed large BC core size and coatings may be due

to the longer westerly transport pathways from sources in these air masses. However, the BC core size was significantly higher in winter than in summer during periods when Beijing received air from the southern plateau. This may result from a large contribution of residential heating activities in southern Plateau in winter which were not present in summer

As Fig. 6 shows, the diurnal variation of rBC mass loading in winter showed a strong anti-correlation with the mixing

layer height (MLH), which means the BC emissions in winter were strongly diluted or concentrated by the development or shrinking of PBL in the daytime and nighttime respectively, but the enhancement in nighttime may also result from the increased emission from heating activities. In summer, the night-time peak of rBC mass loading was absent but peaked during the morning rush hour which may reflect the important contribution from traffic. There were no obvious diurnal variations in either the BC core size or coatings, suggesting BC from different sources was

well mixed during both seasons. In general, larger variations in the physical properties of BC were observed in winter compared to summer.

## 4.2 The size distribution and mixing state of BC

Fig. 7a gives examples of BC core size distributions for typical periods in both seasons. The BC core size distribution

could be modelled as a single lognormal distribution:

$$\frac{dM}{dlogD_c} = Ae^{\left(-\frac{log^2(D_c/D_0)}{2log^2\sigma_g}\right)} \qquad (6),$$





where A is the peak concentration, $D_c$ is the BC core size measured by the SP2, $D_0$ is the core MMD, $\sigma_g$ is the geometric standard deviation (GSD) for the lognormal distribution. The red lines in Fig. 7a show the lognormal fit to the observations. It is noted that some fraction of the distribution at the larger end is not fitted within the single lognormal distribution, which may require an additional moment of lognormal distribution to be accounted for, as has been shown during previous urban studies (Huang et al., 2011). However, the second lognormal fitting will be subject to large uncertainty due to the saturation of the SP2 detector which has an upper cut off at $D_c$=550nm. The additional rBC mass distribution above 550nm may exist but would require instrument reconfiguration to be fully detected. The two-moment lognormal fitting is thus not performed in this study. The extrapolated rBC mass accounted for 5-8% of the total rBC mass loading which is included for the rBC mass loading reported in this study.

Fig. 7b shows the fitting parameters of BC core size distribution at different levels of rBC mass concentration. The core size was generally increased at higher rBC mass concentration but demonstrated considerable variability ranging between 150-220nm. BC particles were observed to have systematically larger core sizes in winter than in summer at the same rBC mass concentration. In winter, the significant increase of core MMD when rBC mass concentration >5µg m$^{-3}$ suggests possible coagulation processes are taking place at high concentration. The width of the core size distribution $\sigma_g$ in winter showed a decreasing trend at higher rBC mass concentration, consistent with the view that coagulation may occur at high rBC mass concentration reducing the width of the size distribution (Pratsinis, 1988). However, in summer $\sigma_g$ showed an increasing trend with rBC mass concentration, which may result from more diverse source contributions at higher rBC mass concentration. The higher $\sigma_g$ in winter than in summer at the same rBC mass concentration suggests a greater complexity of sources in winter. The core sizes observed in Beijing are significantly larger than those observed in London even when the BC source profile was dominated by wood burning (170nm), which may result from other sources of BC.

Fig. 8 shows the coating content of BC was similar between seasons with $M_{R,bulk}$ 1-2 when the rBC mass concentration <2µg m$^{-3}$. During summer the coating thickness only periodically increased ($M_{R,bulk}$ >2); but in winter the coating significantly increased particularly when the rBC mass concentration >3µg m$^{-3}$, showing a highly variable $M_{R,bulk}$ ranging between 1.5-10 (bulk $D_p/D_c$ ~1.4-2.6). Accordingly, the coated BC size $D_{p,v}$ peaked at 220-310nm when rBC mass concentration<2µg m$^{-3}$ in both seasons, however it reached values as high as 550nm under highly polluted conditions. The large variation of BC mixing state during winter-time when the rBC mass concentration >3µg m$^{-3}$, may reflect the additional primary sources such as the large contribution from residential sources during the cold season. However, secondary processing of the complex source mixtures under highly polluted conditions may also play an important role in increasing the coatings.

## 4.3 BC segregation by size-resolved mixing state

Fig. 9 shows the BC core size-resolved mixing state during typical periods in both seasons, and the results obtained in London are also shown as a reference (Liu et al., 2014b). The BC particles were segregated according to the



discontinuous distribution in $E_{sca}$-$D_c$ during different periods. The criteria used is shown by the thick dashed lines in Fig. 9a. Four modes of BC could be segregated: small BC ($BC_{sm}$) with BC cores smaller than 180nm and coating thicknesses <50nm (assuming a core-shell structure); moderately coated BC ($BC_{mod}$) - moderate coating with coating thicknesses of 50-200nm; thickly coated BC ($BC_{thick}$) with coating thicknesses >200nm, and large uncoated BC

($BC_{lg,uncoat}$) with BC core sizes >180nm and thicknesses <50nm. The contribution of these four modes of BC particles to the total rBC number varied during different periods. In summer, the $BC_{thick}$ only contributed a minor fraction of the total number throughout the experiment. The small, moderately coated and $BC_{lg,uncoat}$ fractions are all present in both seasons. Compared with the results in London (Fig. 9e and f), the $BC_{sm}$ fraction was consistent with traffic influences with small core and thin coatings, and the $BC_{mod}$ was broadly consistent with the wood burning observed

in urban London. It is noted that the $BC_{lg,uncoat}$ were not significant in urban London under the different air masses or source influences observed, with the mass fraction of BC IV <8% and number fraction <3% throughout the experimental period. This means $BC_{lg,uncoat}$ may represent a source which was uniquely present in urban Beijing, however not in the UK or surrounding area.

A further analysis on the segregated BC core and coated size distribution (Fig. S2) shows the total size distribution

of BC core and coated size distribution could be generally separated as three lognormal distributions with BC I+IV, BC II and BC III, representing the thin, moderate and $BC_{thick}$ respectively. This in turn suggests the apportioned BC modes may represent discernible primary sources or sources under secondary processing.

**4.4 Comparison of BC source estimation**

Fig. 10 shows the temporal evolution of rBC mass determined by the SP2 and SP-AMS categorized according to the different source contributions derived in section 4.3 and as explored in Wang et al, 2018 respectively. PMF analysis on SP-AMS detected mass spectra identified four factors for BC-containing particles (Wang et al., 2018): fossil fuel BC (FOA_BC) containing BC from vehicle sources and coal burning; BC coated with biomass burning organics (BBOA_BC); BC associated with less-volatile organic coatings (OOA1_BC); and BC associated with semi-volatile

organic coatings (OOA2_BC). Note that each PMF factor includes the refractory BC ($C_x$) and the non-refractory coatings associated with it. In order to directly compare with the rBC mass measured by the SP2, only the mass of the $C_x$ fragments in the PMF factors are used. Table 1 shows the correlation coefficient between the different PMF factors and the SP2 segregated BC types, with green shading highlighting the high correlation ($r^2$>0.6).

The FOA_BC was not able to be further apportioned via the PMF analysis, and so this factor contains both mobile

sources such as from diesel or gasoline engines, and importantly the coal burning emissions. Coal burning could result from both the residential and industrial sectors (Finkelman and Tian, 2018), with the former sector overwhelmingly dominating in winter but the contribution from the latter maintained throughout the year (Fig. S1). The small BC particle component is shown to be solely correlated with FOA_BC ($r^2$=0.68), but FOA_BC is also correlated with moderately coated and $BC_{lg,uncoat}$. The multiple correlations of FOA_BC with BC at different core



sizes and coating thicknesses means the fossil fuel related BC could exhibit a range of mixing states, however the $BC_{sm}$ particle fraction that has smaller core sizes and thinner coatings tends to be only associated with and contribute to the fossil fuel BC fraction.

BBOA_BC which mostly resulted from open cooking sources in Beijing (He et al., 2010), is tightly correlated with $BC_{mod}$ and the large, uncoated BC particle fraction from the SP2. Both BC types had high correlation with FOA and

BBOA since the fossil fuel (excluding the part correlated with $BC_{sm}$) and biomass burning BC particles have similar core sizes and coating contents. The potential contribution from coal burning sources are further investigated by correlating the SP-AMS measured PAH with the SP2-segaragated BC types, as the PAH is considered to be an ideal marker for coal burning (Xu et al., 2006;Sun et al., 2016a). The moderately coated and $BC_{lg,uncoat}$ fractions are found to have the highest correlation with PAH. This means the BC from coal burning tended to contribute to both the

moderately coated and the $BC_{lg,uncoat}$ fractions, but not the $BC_{sm}$, which in turn indicates that the $BC_{sm}$ is mainly a result of mobile sources (such as traffic) rather than coal burning. Previous studies found that the coal emission in urban China may have a larger BC core size compared to traffic sources (Wang et al., 2016). This is also consistent with the lack of the $BC_{lg,uncoat}$ mode in urban London where there was no coal burning present in the city (Fig. 9). Given biomass burning was also correlated with moderately coated and $BC_{lg,uncoat}$ fractions, analyses limited to the

size-resolved mixing state alone may not be able to distinguish the BC particles derived from coal and biomass burning in urban Beijing. This may be because the contribution of biomass burning to BC was significantly lower than that from fossil fuel in Beijing (Zhang et al., 2017), and the open biomass burning was only sporadically significant during spring and autumn harvest time over the NCP region (Chen et al., 2017). In addition, even some of the coal burning contribution, especially for the coal from residential use may be attributed as biomass burning if

using levoglucosan as a marker (Yan et al., 2018), which may lead to some fraction of BBOA identified by the SP-AMS containing some fraction of coal burning. $BC_{mod}$ is consistent with wood burning in urban London but $BC_{lg,uncoat}$ are not present from these sources (Fig. 9e and f). It is therefore more likely that the $BC_{lg,uncoat}$ was mainly contributed by coal combustion.

The thickly coated or $BC_{mod}$ fraction is only well correlated with the BC coated with less-volatile organics

(OOA1_BC) or semi-volatile organics (OOA2_BC) respectively. This means the coating composition would mostly contain less-volatile organic species when BC was thickly coated, whereas the moderate coating BC particle fraction mainly contained semi-volatile species. The coal burning or biomass burning contributions are also significant in the $BC_{mod}$ fraction, which means these primary sources may also emit considerable semi-volatile species internally mixed with BC.


## 4.5 Diurnal variation of different types of BC

Fig. 11 shows that the diurnal variation of the four different rBC types classified by the SP2. In winter all BC types followed the diurnal evolution of MLH, whereas in summer all BC types exhibited a morning rush hour peak. For



BC$_{thick}$ in winter, the average was significantly higher than the median, indicating the sporadic occurrence of the
thickly coated BC. There was no discernible difference in the diurnal patterns of the absolute mass loadings of the
different BC types, which means similar emission sources and the PBL development may have controlled the diurnal
pattern of different BC types to a similar extent. Nevertheless, differences could be identified by relative abundances
of different BC types as discussed in the following section.

The diurnal variation of the number or mass fraction of BC$_{sm}$ (Fig. 11c) peaked at 8am and 7am in the winter and
summer respectively, which corresponded with the morning rush hour. This is consistent with the identified possible
traffic contribution to the BC$_{sm}$ fraction by comparison with the SP-AMS factors (section 4.4). The diurnal variation
of this BC mode also had high correlation with NO$_x$ in both seasons, with the morning rush hour occurring slightly
later in winter than in summer, which further confirms the likely origin of traffic source. The correlation between the
NO$_x$ concentration and different BC types is further evaluated by a multi-linear regression function, expressed as
Equation (7). The fitting parameters are summarized in Table 2.

[NO$_x$]=a0+a1*[small BC]+ a2*[moderately coated BC]+ a3*[thickly coated BC]+ a4*[uncoated large BC]     (7)

Among all BC types, a1 shows the highest value in both seasons, which indicates the strong correlation between the
BC$_{sm}$ mass fraction and the NO$_x$ concentration, with almost identical emission factor of 68-72 ppbv NO$_x$/ μg m$^{-3}$ of
BC$_{sm}$. In winter, the BC$_{mod}$ also contributed some fraction of NO$_x$, whereas in summer the contribution of this BC
type to NO$_x$ was almost negligible. The BC$_{lg,uncoat}$ was not correlated with NO$_x$ emission, which in turn suggests the
coal combustion may not emit significant NO$_x$ (Xu et al., 2000). The BC$_{sm}$ fraction contributed the most in summer
(50-60% in number and 40% in mass) and the second most in winter (30-40% in number and 30% in mass). These
BC particles had smaller core size and thin coatings, and thus they contributed more significantly to the number than
the mass. A similar fraction of the BC was also present in urban London (Liu et al., 2014b) and Paris (Laborde et al.,
2013) and had also been identified to be dominated by traffic sources.

BC$_{mod}$ showed comparable contribution in both seasons (40-50% in mass or 20-25% in number). The fraction of these
particles slightly increased throughout the afternoon for both seasons, and this may partly result from daytime
photochemical processing, though this BC type was also significantly associated with primary sources (section 4.4).
The BC$_{thick}$ fraction showed no apparent diurnal pattern (note that BC$_{thick}$ mass fraction in summer was minor ~5%)
and only made a significant contribution at higher pollution levels (section 4.6).

BC$_{lg,uncoat}$ contributed a significant mass fraction (30-40% in winter and 40% in summer) but contributed little to the
number (<10%) because of the large core sizes. The mass fraction of these BC particles had a pronounced night-time
peak in winter, consistent with the view that this BC type may be contributed by coal burning (also identified by
comparing with the PAH factor in section 4.4), because in the cold season there was significant residential heating
activities at night which may use coal as fuel (Chen et al., 2006). This night-time peak in the mass fraction of BC$_{lg,uncoat}$
was missing in summer due to lack of these heating activities. Nevertheless, the BC$_{lg,uncoat}$ mass fraction remained
substantial in summer, comparable with that of the BC$_{sm}$ fraction at around 40%. This may be because the coal




consumption from the industrial sector that is maintained throughout the year (Fig. S1).

To account for the dilution effect resulting from the development of the PBL, the rBC mass loading was multiplied
by the MLH for every half hour throughout the day. The MLH-corrected rBC mass for each BC type is shown in
Fig. 11b and provides a way of assessing the influence of BC emissions, although there are uncertainties associated
with the MLH determination from ceilometer measurement (Kotthaus and Grimmond, 2018). All BC types showed
a peak in the MLH corrected concentration at night in winter but showed a night-time minimum in summer, which
may reflect a generally higher emission during night-time in winter but not in the summer. During the daytime
between the hours of 6:00 and19:00, $BC_{sm}$ mass corrected for the MLH was similar in both seasons, suggesting that
$BC_{sm}$ may have comparable emission rates between seasons, consistent with the view that these BC may be dominated
by transportation sector (section 4.4). $BC_{lg,uncoat}$ had significantly higher emission at night in winter than in summer
by a factor of 2.5, also consistent with the higher coal consumption from the residential sector in winter.

### 4.6 BC at different pollution level

Given the complexity of the sources contributing to BC in Beijing, the relative primary source contributions and its
interaction with other aerosol species may vary with the overall level of pollution and this in turn may change both
the mixing state and the optical properties of BC. Fig. 12 shows mass fractions of different BC types at different $PM_1$
level determined by the total mass of AMS+SP2. The traffic-like $BC_{sm}$ (Fig. 12a) were a constant fraction of the total
mass at lower pollution levels (when PM1<50 µg m$^{-3}$) that was around 30% and 40% in winter and summer
respectively. The decreased mass fraction of $BC_{sm}$ at higher pollution levels is particularly marked in winter and is
because of the increased contribution of coated BC. The $BC_{lg,uncoat}$ fraction (Fig. 12b) was similar in magnitude to the
$BC_{sm}$ fraction in summer across all levels of pollution, which means the coal-burning like BC was almost as important
as the traffic source. In winter, the contribution of $BC_{lg,uncoat}$ mass was slightly higher than the traffic-like $BC_{sm}$ mass
fraction whereas in summer the $BC_{sm}$ mass was more significant. At the higher pollution levels in winter, some of
the $BC_{lg,uncoat}$ may be also contributed by coagulation.

The fractional mass of $BC_{thick}$ increases substantially at higher pollution levels (Fig. 12d), especially when
PM1>100µg m$^{-3}$ with rBC mass loading > ~2 µg m$^{-3}$ and may be up to 50% of the total rBC mass. The coatings on
these $BC_{thick}$ were largely contributed by secondary species according to the SP-AMS analysis (section 4.4). In
summer, lower rBC mass loadings were observed and correspondingly the $BC_{thick}$ mass fraction was less than 10%.
The mass contribution of moderately or $BC_{thick}$ in summer was lower than in winter at the same pollution level, which
may be due to the higher ambient temperature in summer reducing the amount of semi-volatile material partitioning
to the particles. In both seasons, the mass fraction of $BC_{mod}$ increased with increasing PM1 up to ~40 µg m$^{-3}$, and
slightly decreased at PM1~50 µg m$^{-3}$ due to an increased fraction of $BC_{sm}$ mass. At higher $PM_1$ mass loadings in
winter, the increased mass fraction of $BC_{thick}$ led to a decrease of moderately coated BC. However, in summer the



$BC_{mod}$ fraction increased when PM1>50 µg m$^{-3}$, this high pollution event has previously been shown to be dominated by secondary species (Sun et al., 2015). The higher fraction of $BC_{mod}$ in winter than in summer at the same pollution level may result from greater primary emissions (i.e. more residential coal burning in cold season) or more condensable semi-volatile species at colder temperatures.

The contribution of absorption coefficient is calculated based on single particle information shown in Fig. S4. It is noted that different BC types have different absorption contribution (Fig. 12 right axis) because of their varying absorption efficiency. Fig. S5 shows that the histogram of occurrence for MAC$^{550}$ of each BC type during the experimental period: the $BC_{sm}$ and largely uncoated BC had average MAC about 7.3 m$^2$g$^{-1}$ and 5.3 m$^2$g$^{-1}$ respectively, and $BC_{lg,uncoat}$ had a lower MAC because of its larger core size. The moderately and $BC_{thick}$ had average MAC of 11.2
m$^2$g$^{-1}$ and 12.4 m$^2$g$^{-1}$ respectively.

Fig. 13 shows both modelled and measured MAC$^{550}$ at different pollution levels, the results agree to within 15%. For both seasons, when PM1<50 µg m$^{-3}$, the absorption efficiency of BC only increased slightly at MAC$^{550}$ ~7.8 m$^2$g$^{-1}$ and 7.5 m$^2$g$^{-1}$ in winter and summer respectively. The MAC$^{550}$ for uncoated BC was calculated to be ~6.5 m$^2$g$^{-1}$ for both seasons so the enhancement of absorption efficiency ($E_{abs}$) is calculated as the modelled MAC$^{550}$ normalized by
6.5 m$^2$g$^{-1}$. $E_{abs}$ significantly increased at PM1>50 µg m$^{-3}$ up to 1.5 and 1.9 for summer and winter respectively. There was a wide variability in MAC or $E_{abs}$ at PM1 concentrations of between 100 and 200 µg m$^{-3}$. The generally increase in absorbing capacity of BC at higher pollution level is consistent with the findings of a recent study (Zhang et al., 2018b). The MAC$^{550}$ slightly decreased at very high PM1, i.e. >300 µg m$^{-3}$, and this decrease is more pronounced for measurement (the grey line shows) than Mie-based modelling. This may result from the shadowing effect that the
very thick coating may shield the incident photons onto the absorbing core (He et al., 2015;Zhang et al., 2018a).

## 5 Conclusion

In order to probe the sources and processes governing atmospheric black carbon in Beijing, measurements were performed in both winter and summer using an SP2 and the core size and coating thickness were examined using the
single particle data. Higher rBC mass loading with more variable coatings was found in winter than in summer. The air mass from the Southern Plateau brought BC with the largest core size and coatings in winter, indicating the appreciable regional influence, whereas in summer the characteristics of BC were relatively independent of air mass direction. In contrast to equivalent measurements in London, where two particle types were observed corresponding to traffic and wood burning, other types were observed in Beijing, probably reflecting a more complex mixture of
sources. rBC number and mass concentrations were quantified according to the following four particle types: small thinly-coated BC, moderately-coated BC, thickly-coated BC and large thinly-coated BC. By comparison with other measurements, in particular a factorisation of coating materials from the SP-AMS, these were assigned to different soot sources.





The small thinly-coated rBC fraction was associated with traffic emissions and made up for 30% and 40% of the rBC mass in winter and summer respectively. This particle fraction was strongly associated with $NO_x$, though the implied ratio of 14 ng m$^{-3}$ ppb$^{-1}$ was lower than the values of 18-28 ng m$^{-3}$ ppb$^{-1}$ reported for London, likely due to differences in the emissions fleet, such as a more widely used gasoline engine in Beijing (Wang et al., 2009). The large thinly-coated rBC could be associated with coal combustion and corresponded to around 30-40% and 40% of the rBC mass in winter and summer respectively.

The moderately-coated particle fraction made up 40-50% of the rBC mass and was associated with both emissions and atmospheric processes. As a result, the original source of these particles is currently ambiguous; it is possible that this class has multiple contributions. The thickly-coated particle fraction was mainly present during the winter heavy haze events when PM1 was greater than 200 µg m$^{-3}$ or the rBC mass loading was greater than 4 µg m$^{-3}$. During these events, these particles made up for around 20-45% of the rBC mass and the coatings could be associated with 495 secondary species, implying that these are rBC particles that have undergone some form of atmospheric processing. Given the thick coatings, it would be expected that these particles would exhibit higher mass absorption and scattering coefficients, higher hygroscopicities (thus high optical thickness in the upper boundary layer) and greater scavenging potential (thus shorter atmospheric lifetime). These large BC importantly enhanced the absorption efficiency at high pollution levels, and reduction of emissions of these large BC particles and the precursors of the associated secondary 500 coating will be an effective way of mitigating the heating effect of BC in urban environments.

**Data availability**

Processed data is available through the APHH project archive at the Centre for Environmental Data Analysis (http://data.ceda.ac.uk/badc/aphh/data/beijing/). Raw data is archived at the University of Manchester and is available 505 on request.

**Author contributions**

D.L., J.D.A., M.J.F designed the research; J.D.A., D.L. R.J. J.W. C.Y. J.D.A. H.C. M.J.F, C.X., J.L. F.S. X.G. Y.S. and P.F. performed experiments; D.L. performed the data analysis; J.W. and X.G. analysed the SP-AMS data; S.K. and S.G. analysed the MLH data; J.L. and F.S. analysed the $NO_x$ and CO data; D.L., J.W. J.D.A., and H.C. wrote the 510 paper.

**Acknowledgments**

This work was supported through the UK Natural Environment Research Council (grant refs. NE/N007123/1, NE/N00695X/1, NE/N00700X/1), the National Natural Science Foundation of China (41571130024, 41571130034, 515 21777073).





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





| $R^2$ (N=1406) | FOA_BC (fossil fuel) | BBOA_BC (biomass burning) | OOA1_BC (less volatile) | OOA2_BC (semi-volatile) | PAH_BC |
|---|---|---|---|---|---|
| Small BC | 0.68 | 0.55 | 0.10 | 0.12 | 0.47 |
| Moderately coated BC | 0.61 | 0.66 | 0.40 | 0.63 | 0.81 |
| Thickly coated BC | 0.31 | 0.54 | 0.61 | 0.47 | 0.59 |
| Large uncoated BC | 0.79 | 0.85 | 0.42 | 0.34 | 0.82 |
| BCmass total | 0.74 | 0.85 | 0.51 | 0.51 | 0.89 |


Table 1. (a) Pearson's correlation coefficients between the time series of SP2 and SP-AMS segregated rBC mass and PAH. The correlations with Pearson $r^2 > 0.6$ are shaded in green. All correlations are significant at the 0.01 level (2-tailed).


| NO$_x$ with BC types | a1 | a2 | a3 | a4 | a0 |
|---|---|---|---|---|---|
| winter | 72.3 ± 3.2 | 21.3 ± 3.0 | 16.3 ± 2.1 | 0 | 8.8 ± 1.3 |
| summer | 68.4 ± 4.7 | 0 | 0 | 0 | 1.15 ± 0.77 |

Table 2. Multi-linear regression for Equation (7) among NO$_x$ and different BC types.



(a)

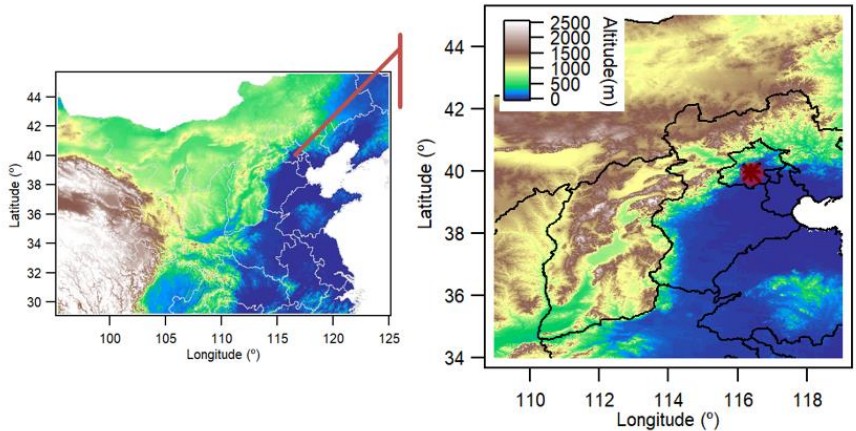

(b)                                    (c)

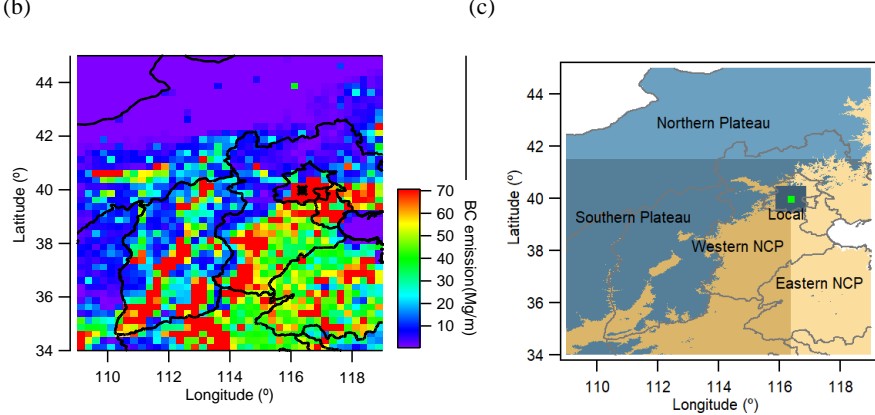

Figure 1. (a) Terrain height for the North China Plain (NCP) and the Plateau; (b) BC emissions from all sectors in
2010; (c) the regional classification according to terrain height and emission.

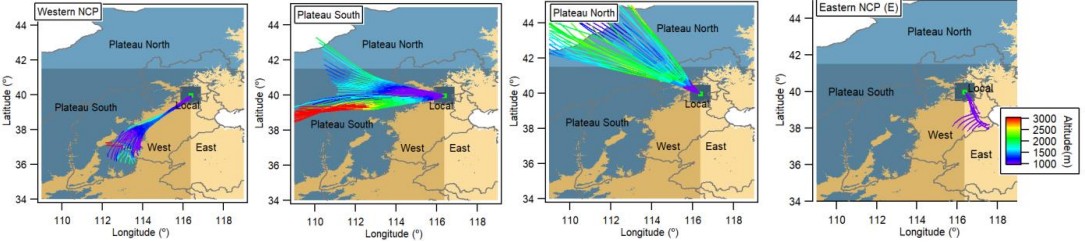

Figure 2. Classified air mass origins based on back trajectory analysis: western NCP, Southern Plateau, Northern
Plateau and Eastern NCP.






(a)

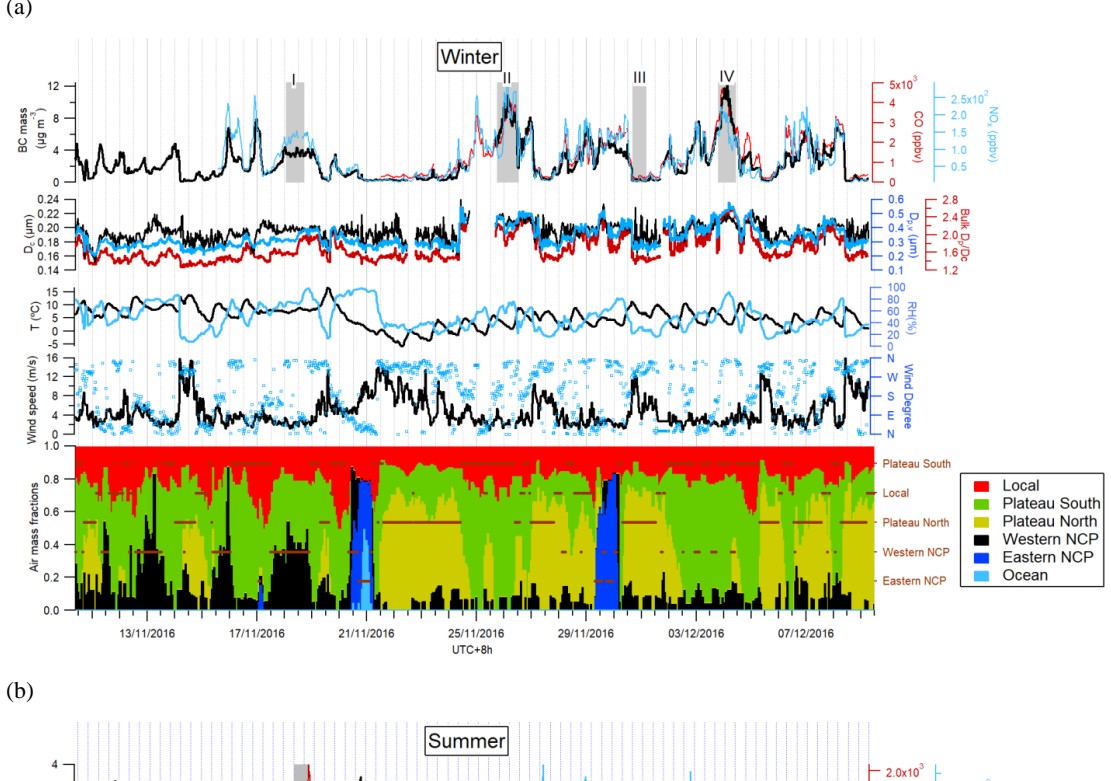

(b)

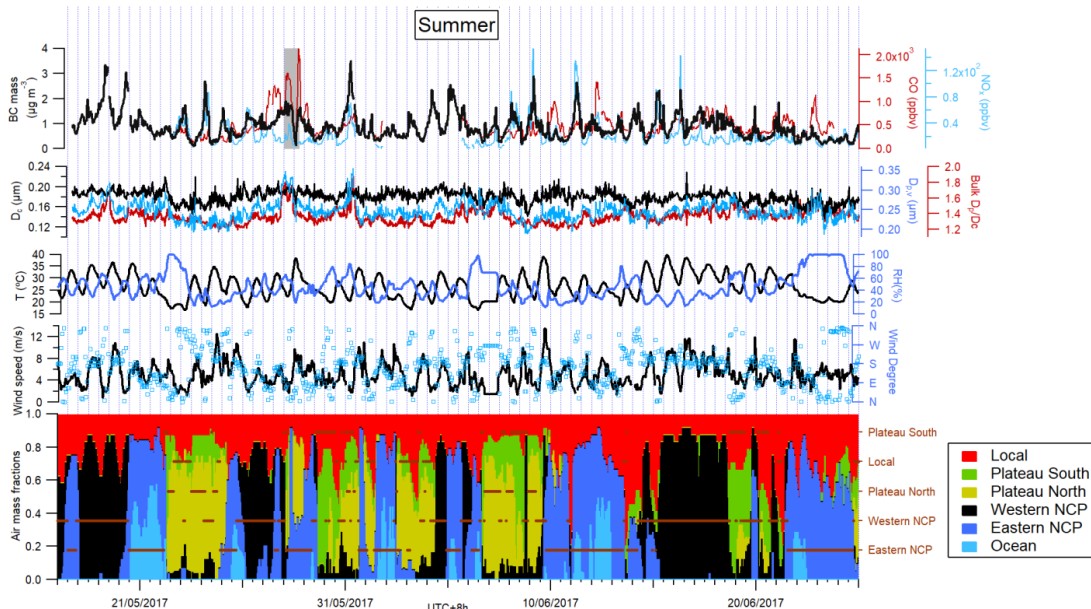

Figure 3. Time series of BC-related properties in the winter (a) and summer (b) experiments. From top to bottom
subpanels: BC mass loading, CO and $NO_x$; BC bulk $D_p/D_c$, core MMD and coated volume-equivalent diameter;
ambient temperature and RH at z=8 m; wind speed and direction at z=120 m; air mass fractions from different origins





based on back trajectory analysis, with the brown horizontal lines indicating the classified air mass types. The vertical grey shades mark the periods for the detailed mixing state analysis in Fig. 9.


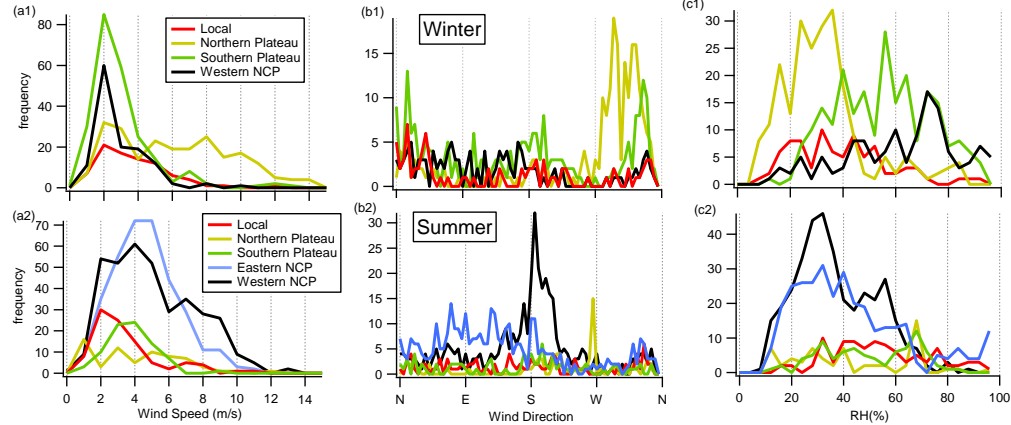

Figure 4. Frequency distributions of wind speed and direction (z=120 m) and RH (z=8 m) for classified air mass types in both seasons.

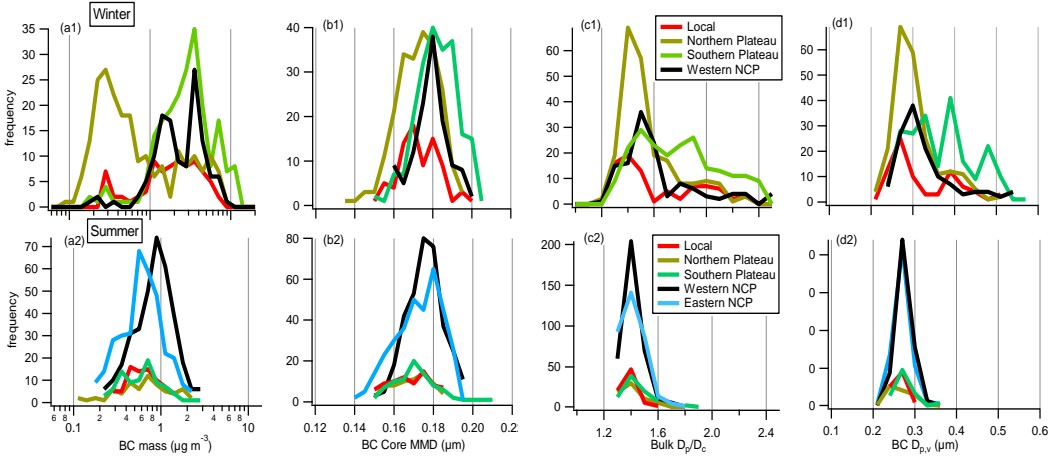

Figure 5. Histograms of BC mass loading, core size MMD, $D_p/D_c$ and $D_{p,v}$ for different air masses in both seasons.





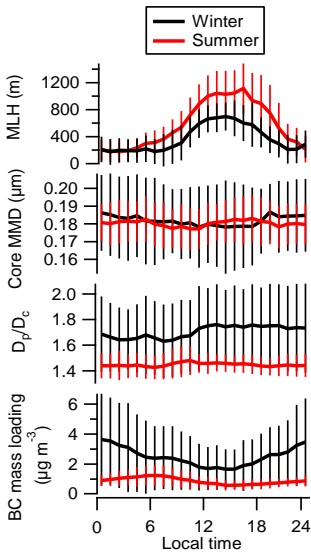

Figure 6. Diurnal variations of mixing layer height (MLH) and BC-related properties in winter and summer. The
lines show the mean at each hour and error bars denote ±1σ.

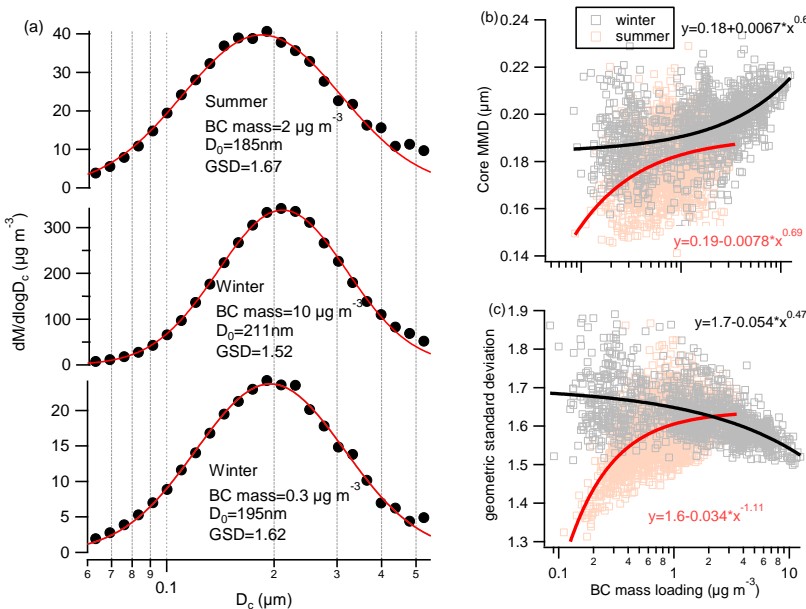

Figure. 7. (a) BC core size distribution averaged over different rBC mass loading conditions in both seasons,
showing the lognormal fitting and the fitted peak diameter ($D_0$) and geometric standard deviation (GSD); (b) BC
core MMD as a function of BC mass loading and parameterization; (c) BC core GSD (from the fitting) as a
function of BC mass loading and their fitting functions.




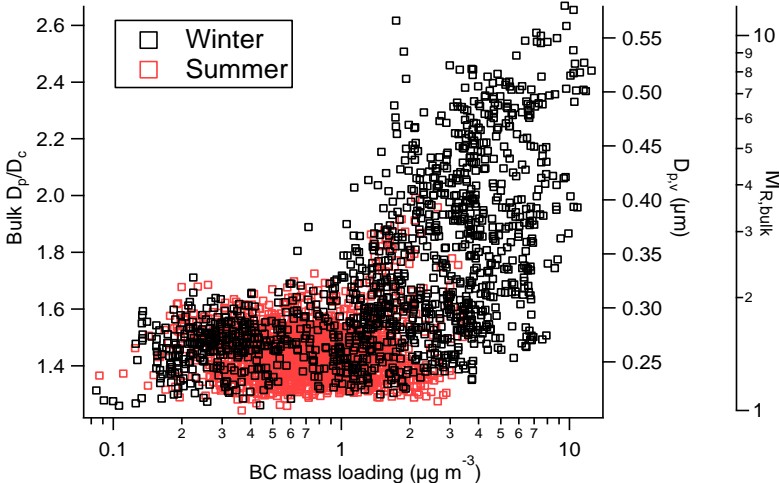

Figure 8. Mixing status of BC at different levels of BC mass loading in both seasons, with different y-axes showing
bulk $D_p/D_c$, volume-weighted coated particle size ($D_{p,v}$) and bulk mass mixing ratio of coating/rBC.


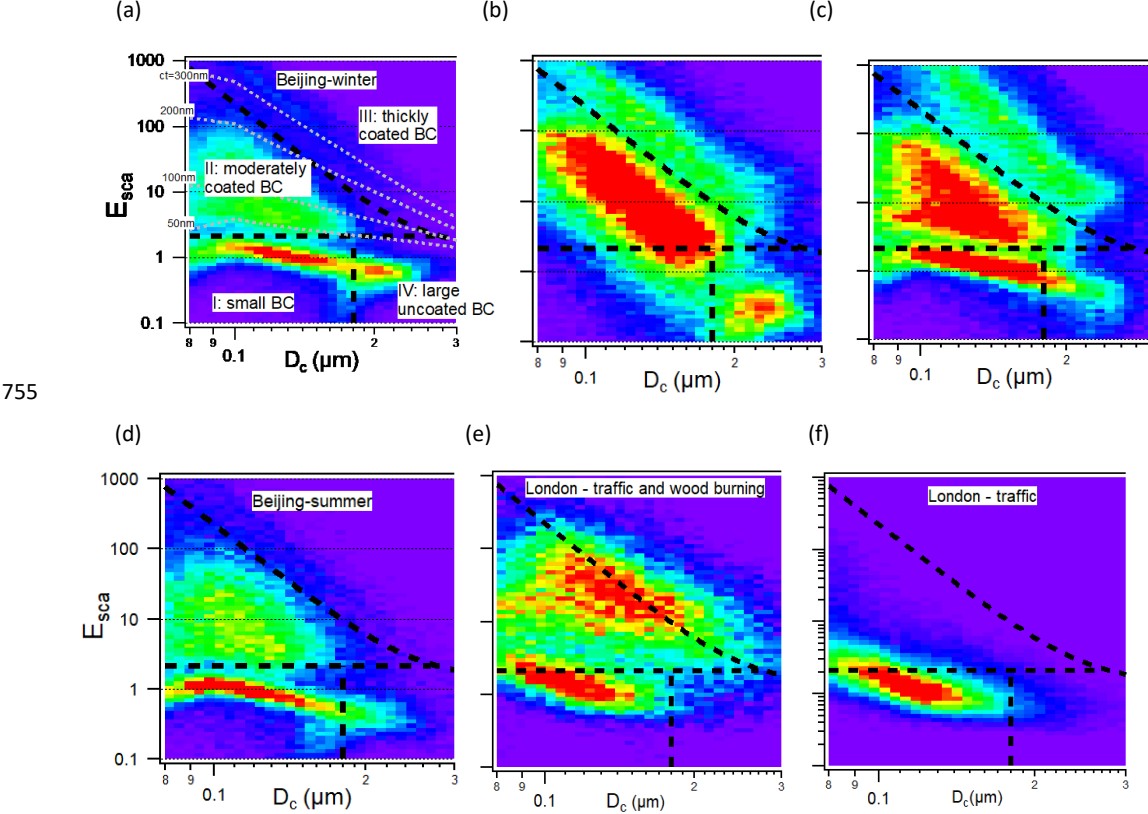


Figure 9. Scattering enhancement ($E_{sca}$) as a function of BC core size ($D_c$) for the three periods (as period I-III indicated in Fig. 3) in Beijing winter (a-c), Beijing summer (d), London with mixed sources (e) and London with traffic source (f). Each plot is coloured by particle number density. The particles are separated as four groups using the borders (from top to bottom) at $y=3.38+0.000436 \cdot x^{-5.7}$, $y=2.1$, $x=0.18$, as shown by dashed lines on each plot. The grey dashed lines on (a) denote coating thicknesses mapped on $E_{sca}$-$D_c$ plot.






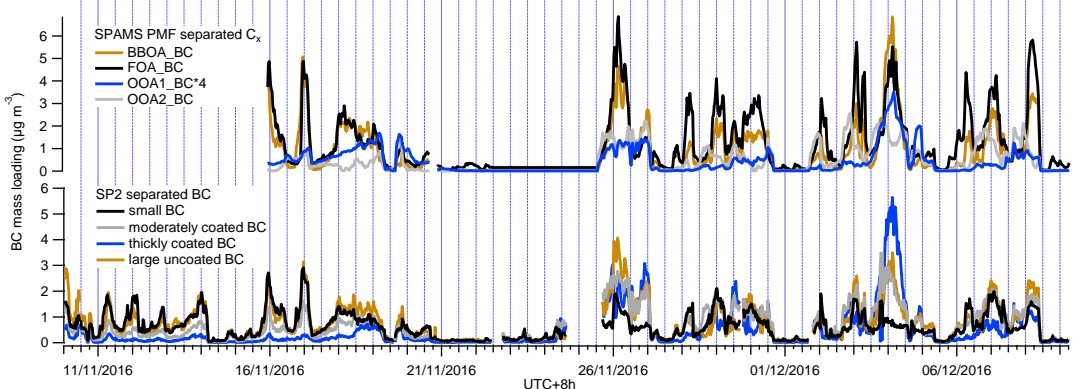

Figure 10. Time series of SP2-separated rBC mass and SP-AMS separated $C_x$ mass by PMF analysis.

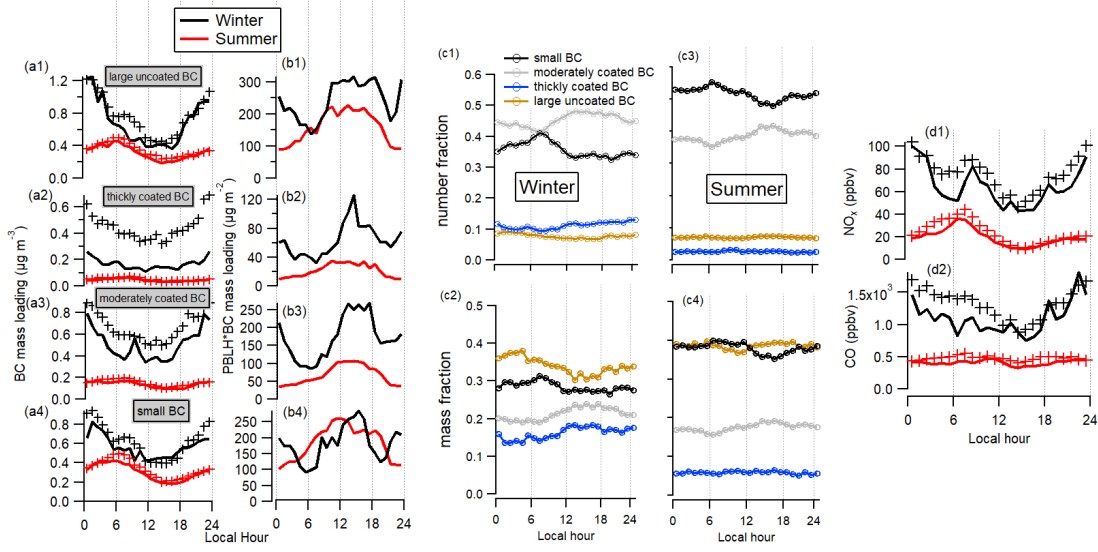

Figure 11. Diurnal variation of the rBC mass segregated by their SP2 characteristics in both seasons. a) shows the median (solid line) and mean (markers) at each hour; b) the median value of PBLH-corrected rBC mass; c) the

number and mass fractions for each BC type; d) the $NO_x$ and CO.



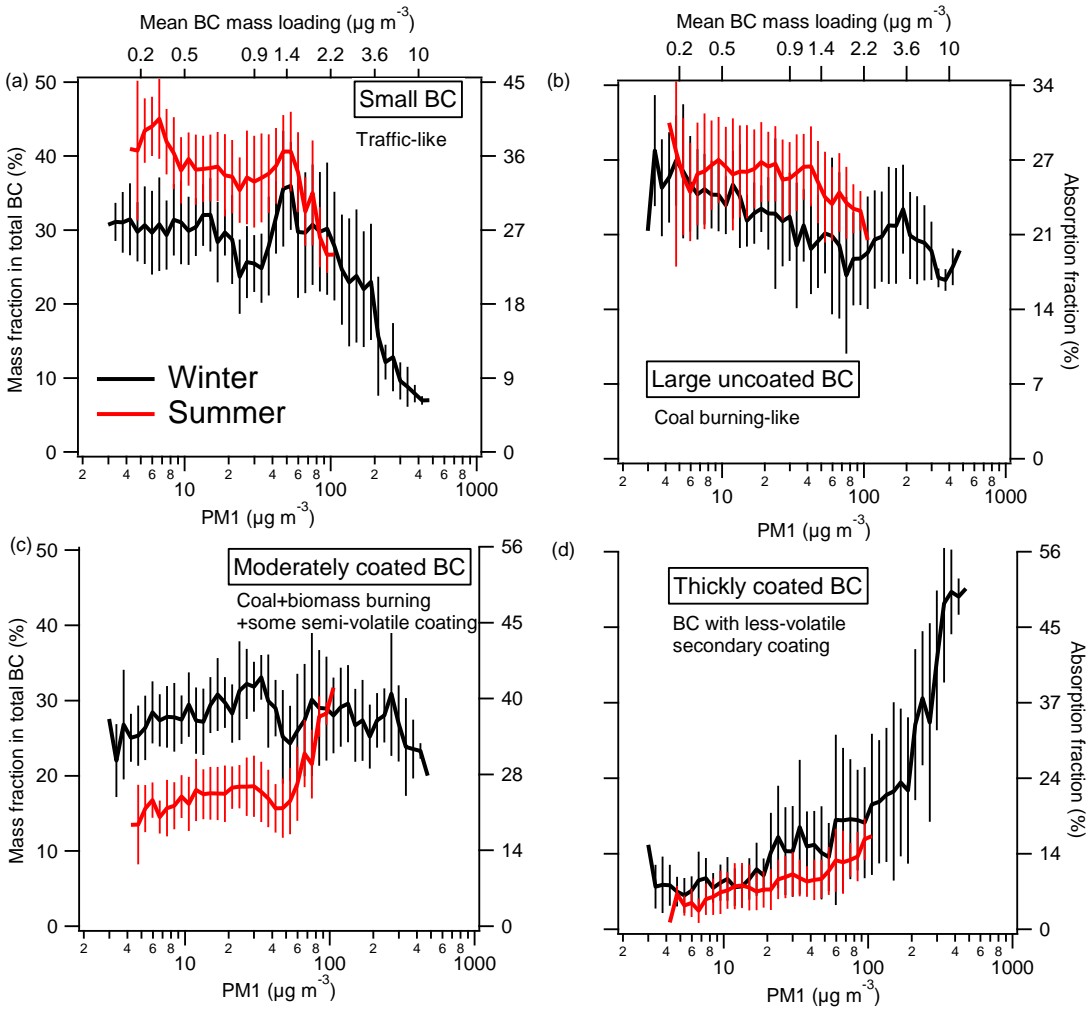

Fig. 12. The mass (left y-axis) and absorption (right y-axis) contributions of different BC types at different pollution
levels. The mass fraction of each BC type is the average ± standard deviation at each PM$_1$ bin. The top axis shows
the averaged rBC mass loading at each PM$_1$ bin, and the right axis indicates the average absorption fraction
corresponding with each rBC mass fraction.




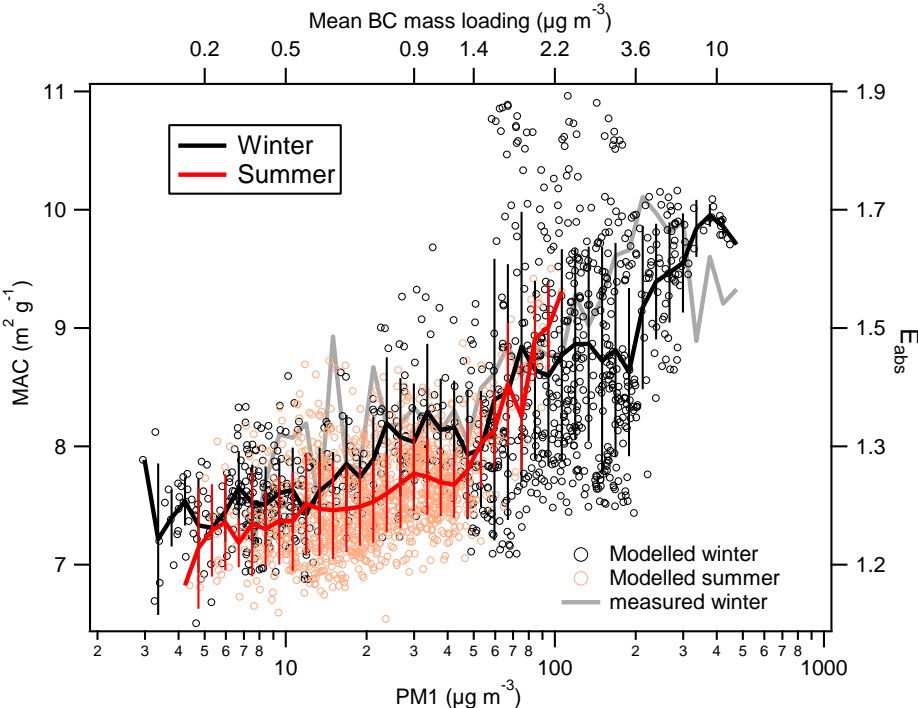

Fig. 13. The modelled and measured MAC$^{550}$ and $E_{abs}$ at different pollution level.