# Peer review of "Contrasting physical properties of black carbon in urban Beijing between winter and summer"

_Atmospheric Chemistry and Physics, 2018_

## Referee Comment (RC1) · Anonymous Referee #1 · 10 Dec 2018

Review of acp-2018-1142
Liu et al. "Contrasting physical properties of black carbon in urban Beijing between winter and summer"

Summary:
The authors present a study investigating mixing state and composition of black carbon (BC)-containing particles collected in Beijing over two seasons. They have included comprehensive measurements and/or calculations of the physical, optical, and chemical properties associated with these particles. They utilize this information to provide detailed discussion about the sources of these BC-containing particles and how different sources yield different aerosol properties (Figure 9 is especially nice in this regard!). I recommend that this manuscript be promoted from discussion paper to publication in *Atmospheric Chemistry and Physics* once the authors have sufficiently addressed comments from all reviewers.

General comments:
- The authors appear to be "loose" with statistics throughout the manuscript. For example, in lines 258, 266, 294, and 384 (and potentially elsewhere), differences between properties etc. are described as "significant" or "not significant", but no statistical metric indicative of significance testing (e.g., *p*-value) has been reported. Moreover, in lines 269 and 392 (and again, potentially elsewhere), reference is made to correlation/anti-correlation, but there is no metric reported. I suggest that if the authors would like to use this language, they should support this with some metric.
- Section 3.1: The authors appear to move back and forth between calculations representing all BC particles over a given time window (e.g., mass median diameter, volume-weighted coated BC size) and calculations for single particles (e.g., scattering enhancement, mass absorption cross section). Consequently, I found this section challenging to read. Could the authors please re-write this section for clarity?
- Section 4.6: The authors switch back and forth between symbolic and text representations for the different BC types (e.g., Line 446, 458, 459), which is a little distracting. Please correct for consistency one way or the other.
- Something that is not clear to me is how the authors "combine[d] two online source apportionment methods". Is this the point of Figure 10? Given that this is a key highlight in the abstract, the demonstration of this in the manuscript is weak (or at best, under-emphasized). The authors should address this concern with their revisions.

Specific comments:
- Lines 155-157: I am not questioning the authors' approach (i.e., the "why"), but I don't quite follow the "what". Could the authors please clarify this? The sentence beginning with "the fraction of 10%" is especially confusing to me.
- Line 171: I believe this should be "$cm^{-3}$"
- In 224: In my experience, it is "low-volatility" or "lower-volatility" organics, rather than "less-volatile" organics. (If the authors want to keep their terminology based on what they are more used to, this is fine)
- Lines 264-265: Does distance or time matter more? A longer transport pathway does not necessarily mean that something is more aged. Also, I thought that all of the back trajectories were limited to 24 hours.
- Lines 337-340: These source terms were defined previously but without the "BC". Please update the manuscript for consistency.
- Lines 368-371: Not a comment to address, but I learned that levoglucosan can be present in coal combustion!

- Line 396: How well does this linear model work? I'm not seeing anything demonstrating the quality of fit (e.g., $R^2$) or even a plot.
- Figure 6: I suggest that the authors "jitter" either winter or summer to make the error bars clearer (e.g., shift one of them ½ hour to the left or right along with a note in the caption)

---

## Referee Comment (RC2) · Anonymous Referee #2 · 18 Dec 2018

Overall, I find this to be a largely observational paper that provides some interpretation. I find it to be written in such a way that it is often difficult to follow. Most often this is because of a too-rapid going back and forth between winter/summer. I think that the authors should strongly consider trying to organize each section to fully describe each season, and then make comparisons. This would also really help in instances when they are trying to make specific points about specific seasons. I do also have some concerns about what the Esca-Dc plots mean when the Esca is < 1, and what this says about the uncertainty of the method overall. I agree with the first reviewer that the "what" is generally (although not always) clear, but the "why" is often lacking. I think the authors could do a better job at supporting their conclusions. There is a lot of very specific terminology used throughout and a summary table would be very

helpful/welcome. I think that the measurements are, overall, of good quality and the analysis is likely robust, and thus the manuscript could ultimately be publishable. However, I do also think that the manuscript could do with some structural reorganization within sections, with some greater details, and with stronger connections between observations and interpretation. My specific comments follow below, with ** put next to those that I think are more crucial.

L34: The meaning of "dilution effect" is not clear.

L36: The meaning of this sentence about source apportionment methods is not clear. What does "physical method" mean? And how are these combined if they are, apparently, performed separately? (Perhaps the text addresses this, but the abstract is unclear.)

L42: What does "tended to dominate with moderate coatings" mean? These particles do not dominate the BC mass overall. Are words missing?

L100: The use of the word "novel" does not seem appropriate here. The techniques have been used previously. Perhaps a "novel combination" is appropriate, but even then I'm not certain as there are other studies that have looked at BC size distributions and composition.

References Liu et al. (2014a) and (2014b) are the same reference.

Fig. 1: The units on the emissions are not clear. It says Mg/m. Why per meter?

L157: It is unclear how the authors established that this "is the optimum metric to reflect…" In what way specifically was it optimum? How was this established specifically? How is varying by +/-10% the right value?

L181: What is the smallest coating amount that can be reliably determined, given the uncertainty of the method?

L195: I suggest the authors use a sub, rather than superscript for the MAC, such that

there is zero ambiguity as to whether this is an exponent.

L205: How was the PAX calibrated? How do the MAC values measured at 870 nm compare with those calculated at 550 nm?

L223: What does it mean to apply PMF in "real time"? Per the cited Wang paper, the PMF analysis was conducted using standard methods, which are certainly not applied in "real time".

L233: I am not certain that the statement "As the bottom panels show, the site was mostly influenced by northerly air masses in winter..." is justified by the data. The figure shows that probably half of the total period was dominated by Plateau South air masses.

Fig. 3: The authors should consider using a different color scheme, especially one that does not put green and red next to each other.

L244: It would be helpful if the authors would elaborate on the meaning of the following sentence: "In summer, air masses from the western NCP showed lower RH which may result from the almost latitudinally homogenous distribution of higher temperatures." How does a "homogenous distribution" translate to lower RH? I understand why higher temperatures at the point of measurement might. But higher temperatures elsewhere could, at least in theory, lead to increased evaporation of water.

L250: The authors state that rBC concentrations are higher in winter due to higher emissions. But boundary layers are often also lower in winter, leading to higher concentrations of primary pollutants. How can the authors separate these effects, or at least rule out boundary layer differences as an important reason for the wintertime increase? This should especially be rationalized with the authors statement above that wintertime saw more air masses from the northern plateau yet that the northern plateau airmasses were linked to periods of the lowest concentrations. And since the authors note boundary layer height differences below (L269). They do this below, but

they might either bring this discussion up or point the reader to the later discussion.

L256: When the authors state "This may be also..." the also makes me think that there has been some argument advanced already. But they do not seem to advance an argument before this point as to why the particles sizes from the northern plateau were smaller. Why would lower concentrations mean smaller sizes?

Fig. 5: If the authors were to show the averaged BC size distributions for the different air masses, perhaps as a supplemental figure, this would help the reader to understand the MMD histograms.

L264: It is not clear why "longer westerly transport" would result in larger rBC cores. This aspect needs to be justified.

** Section 4.1: I find that the authors bounce between summer/winter very quickly, and not always clearly. I suggest that this might be clearer if the authors were to fully present one season, and then the other, and then point out notable similarities/differences. As written, I find this more difficult to follow than it need be. This continues through many of the sections.

L270: Just to be clear, when the authors refer to BC being "concentrated" by the shrinking of the PBL at night, they are not implying that the shrinking itself concentrates the BC, correct? This is not physically what happens. Emissions that do occur at night are into a smaller atmospheric region and thus end up more concentrated.

**BC Sources: Is BC from coal expected to be chemically similar as BC from other sources? I could see reasons it might be quite different chemically, and therefore quite different optically. If coal combustion, especially residential coal combustion, is an important source of wintertime BC, could it be possible that the coating amount estimation method might have some trouble, as it uses an RI that was determined for BC from (most likely) vehicle combustion? Is there any evidence available in the literature to illustrate that BC from coal behaves similarly and that the methods applied

are appropriate? Has any direct source testing been done? I am unaware of any SP2 source sampling of coal. Yet, we know that the properties of BC can influence the SP2 measurements (e.g. Laborde et al.)

L274: I am not sure about the statement that there were no "obvious" diurnal variations in the BC coatings. The figure suggests a reasonably evident increase around 10 am in winter and a small increase around 9 am in summer. And in winter the size seems to be notably higher at night than day. The variation is not huge, but it seems evident. The authors use this statement regarding no "obvious" variations to argue that things are "well mixed during both seasons." This seems to be a bit of a stretch.

**L280: Is the conclusion that the size distributions are log normal robust over time? The two winter distributions shown suggest that if a campaign average were calculated one might need to use a multimode fit. Is this conclusion only true for a relatively narrow BC mass concentration range? That the MMD varies with BC concentration (Fig. 7b) suggests that the overall average distribution is not fittable by a single lognormal mode. Further clarification is needed. Also, are the fits given in Fig. 7b/c meaningful? These seem like arbitrarily chosen functions. Given the scatter in the data, one might think another functional form would work nearly as well. Are these just to show that the data vary? The authors might consider binning the data instead to illustrate this point.

**L295: The authors suggest coagulation might be responsible for the increase in BC size when the BC concentration is large. How can they exclude a shift in source? If they are going to speculate about one reason, they should speculate about the other reasonable interpretation (change in source). Also, arguments regarding coagulation would be strengthened if the authors could point to the fraction of the total particles that contain BC. Only BC-BC coagulation leads to growth. BC coagulation with non-BC particles does not lead to growth of the BC core. In many environments, BC is only a small fraction of the total particles. What is the situation here? I suggest this should be discussed. Especially, I don't understand how the authors can argue that a shift in sigma during one season is likely due to changes in source but in another it is

coagulation just because the direction of the shift is different. This requires the authors knowing a priori what the end members of mixing line are with respect to width and MMD. The weakest part here, in my opinion, is in the range of concentrations where the two seasons overlap (0.1-4 ug/m3) and where different behavior is observed. Why would coagulation drive behavior during one season in the overlapping region but not the other? Especially when the authors argue for a "greater complexity of sources" in winter. I suggest this paragraph needs substantial revision.

**L310: The discussion regarding the increase in coatings in the winter at higher BC concentrations would be, in my opinion, greatly strengthened if the authors also considered how the absolute concentrations of other PM species varied. Also, I find this discussion to be very weak in the context of the available data. The related paper (Wang et al.) uses PMF to analyze the coatings on BC. There are primary and secondary coating materials identified. How do these play into things? The discussion, as presented, is just statements of obvious factors that might impact coating amounts. But the authors could, and should, go beyond this, given the available data.

**Fig. 9: How should one interpret the large number of points below the Esca = 1 line? Presumably, no point should be below this line if the interpretation is robust. This is especially important for the "large uncoated BC" region, which is almost entirely in a range where signal should not exist. Also, what is the smallest core size for which a coating can be reliably determined? (Is there any mismatch between the incandescence and scattering lower size detection limits?) I do understand that there is uncertainty in the measurements, and that perhaps this is what contributes. But the particular patterns of the Esca-Dc relationship, which trend towards values of Esca < 1 as Dc increases in general, really make me question the robustness of the method and interpretation.

**Fig. S2: I do not understand this figure. The underlying distributions appear to have arbitrary sizes. I think these are somehow derived from the Esca-Dc relationship. But they are very oddly shaped, essentially unphysical. This figure is mentioned briefly as supporting conclusions regarding sources. But given that the shapes of the underlying

distributions are so very, very strange, I believe that it requires substantial additional discussion. These are most clearly not log normal, as stated by the authors.

L342: This should clarify that these are linear fits.

Section 4.4: It would be helpful to clarify that this is only for winter right at the start of this section.

Section 4.4: The authors need to clearly define what they consider good, moderate, poor, etc. correlations. A value of $R^2 > 0.6$ is stated as both "high" and "moderate" and "tightly" for example. Use of consistent language would facilitate consistent interpretation.

**L355: I am finding it difficult to understand all the terminology. The authors are variously defining things by ranges (I-IV), types (fossil fuel, biomass burning, traffic), etc. An effort to really clarify all the terminology would be most welcome and would facilitate the readers understanding. This is especially true when the authors make statements such as that the fossil fuel and biomass burning have similar core sizes and coating contents. I am having a very difficult time understanding what, specifically, the authors refer to (especially when I look again at Fig. S2, where the distributions of the different types seem to vary quite greatly).

OOA: Per the complementary Wang et al. paper, the OOA2/BC ratio is notably larger than the OOA1/BC ratio. Yet the OOA2_BC factor is more correlated with "moderately" coated BC while the OOA1_BC is more correlated with "thickly" coated BC. Can these be reconciled?

L374: This sentence could be rewritten to make it clearer. Use of commas, at least, would help.

L382: The wintertime BC was, generally, anticorrelated with the MLH. It did not "follow" the MLH.

L401: The citation to Xu et al. (2000) just points out that NOx can be controlled from

coal combustion. It does not address the extent to which modern NOx controls are implemented today in and around Beijing. A more modern reference of direct relevance would be welcome.

L404: Since no large, uncoated BC mode was observed in London, and since this makes up a large fraction of the total, is it fair/relevant to compare the absolute fractions from this study to a study from London? I am not sure that it is. The authors might consider renormalizing, removing the large, uncoated BC mode from the statistics, if they wish to compare in this semi-quantitative way.

Fig. 9: Is each panel individually normalized to the maximum?

\*\*Section 4.6: I am surprised to not see a discussion of how, perhaps, the small, uncoated BC is converted to thickly coated BC when the PM levels increase. It is evident from Fig. 12 that the sharp drop in the fraction of small, uncoated BC results largely from an increase in the thickly coated BC. What I find in this section is largely just a statement (or really, a series of statements) as to how the fractions of one type change with another. But there do not seem to be a lot of insights that actually come out of this section, in my opinion. I suggest that the authors focus more on development of insights rather than just a statement of relationships. Where they do try to develop insights, they really come off as speculative (e.g. L447) rather than fully developed.

Fig. 12/L455: if the calculations of absorption really account for single particle coating state, then I have a difficult time understanding how there is a linear translation between the particle fractional contributions (left axis) and the absorption contribution (right axis). This implies that there is a single, characteristic value for each BC type that the fractional contribution can be multiplied by. But this wouldn't seem to go with the single-particle analysis. Is the single particle analysis not applied at each point in time to understand the variability within the different classes, but instead applied as a class average? I think it is the latter based on the discussion, but this could be clearer.

L461: How was the MAC at 550 nm determined? The measurements were made at

870 nm. I find this unclear. Also, people typically think of BC as having an MAC at 550 nm around 7.5 m2/g (see e.g. Bond et al. (2006)). Yet, the translation between the MAC and Eabs in Fig. 13 implies a smaller MAC was used with the measurements, and this is confirmed on L458, although there is a seemingly contradictory value on L465 (which is perhaps just demonstrating the inadequacy of Mie theory for calculation of MAC values.) Unless what the authors are doing is actually showing the measured Eabs and a modeled MAC. It is not clear what the authors have done here.

**Section 4.6 – Absorption: The authors report their MAC values, but provide very little interpretation. Some interpretation would be welcome. They cite the Zhang (2018b) paper as some support of the reasonableness of their observations. But, Zhang et al. (2018b) find Eabs values at the same wavelength that never fall 1.7 while here the authors find at about the same conditions values of 1.2. (Personally, I think there are substantial problems with the Zhang et al. (2018b) paper, but nonetheless there is an inconsistency that challenges the simple citing of this as support.) The total PM/BC ratio appears to increase with the PM1 concentration. Could there be additional brown carbon leading to the increase? Or do the authors think that the increase results from the coatings? Is this what they are trying to imply (but not stating directly) when they compare the observations to the calculations (although as I note above the origin of the observations at 550 nm is not clear)? I think the authors should be more explicit. Note also that there is only a "shadowing" effect (L470) if the coating is absorbing.

---

## Author Comment (AC1) · 20 Feb 2019

Firstly, we would like to thank both referees for their important comments. We have addressed all of the comments below. The original comments from referees are in normal font, our replies are in blue and the tracked changes in the main texts are in red.

Review of acp-2018-1142
Liu et al. "Contrasting physical properties of black carbon in urban Beijing between winter and summer"

Summary:

The authors present a study investigating mixing state and composition of black carbon (BC)-containing particles collected in Beijing over two seasons. They have included comprehensive measurements and/or calculations of the physical, optical, and chemical properties associated with these particles. They utilize this information to provide detailed discussion about the sources of these BC-containing particles and how different sources yield different aerosol properties (Figure 9 is especially nice in this regard!). I recommend that this manuscript be promoted from discussion paper to publication in *Atmospheric Chemistry and Physics* once the authors have sufficiently addressed comments from all reviewers.
We thank the referee for their positive comments.

General comments:

The authors appear to be "loose" with statistics throughout the manuscript. For example, in lines 258, 266, 294, and 384 (and potentially elsewhere), differences between properties etc. are described as "significant" or "not significant", but no statistical metric indicative of significance testing (e.g., $p$-value) has been reported. Moreover, in lines 269 and 392 (and again, potentially elsewhere), reference is made to correlation/anti-correlation, but there is no metric reported. I suggest that if the authors would like to use this language, they should support this with some metric.
We have now added the significance level (p-value) for each statement when referring to the significance level. In some cases, we have not performed a statistical test. In such cases the word "significant" is removed. We have also added all Pearson $r^2$ value to explain the correlations.

Section 3.1: The authors appear to move back and forth between calculations representing all BC particles over a given time window (e.g., mass median diameter, volume-weighted coated BC size) and calculations for single particles (e.g., scattering enhancement, mass absorption cross section). Consequently, I found this section challenging to read. Could the authors please re-write this section for clarity?
We have rewritten some parts of this section, such as adding "the abbreviations using subscript $i$ refer to the single particle, variables without a subscript refer to the bulk information".

Section 4.6: The authors switch back and forth between symbolic and text representations for the

different BC types (e.g., Line 446, 458, 459), which is a little distracting. Please correct for consistency one way or the other.

We have now made the abbreviations consistent.

Something that is not clear to me is how the authors "combine[d] two online source apportionment methods". Is this the point of Figure 10? Given that this is a key highlight in the abstract, the demonstration of this in the manuscript is weak (or at best, under-emphasized). The authors should address this concern with their revisions.

Thanks to the referee for pointing this out. We should have used "compare" and not "combine". This has been corrected.

Specific comments:

Lines 155-157: I am not questioning the authors' approach (i.e., the "why"), but I don't quite follow the "what". Could the authors please clarify this? The sentence beginning with "the fraction of 10%" is especially confusing to me.

This fraction is arbitrarily determined and we have clarified this in the revised version.

Line 171: I believe this should be "$cm^{-3}$"

Corrected.

In 224: In my experience, it is "low-volatility" or "lower-volatility" organics, rather than "less-volatile" organics. (If the authors want to keep their terminology based on what they are more used to, this is fine)

It has been changed to "low-volatile".

Lines 264-265: Does distance or time matter more? A longer transport pathway does not necessarily mean that something is more aged. Also, I thought that all of the back trajectories were limited to 24 hours.

This sentence has been revised.

Lines 337-340: These source terms were defined previously but without the "BC". Please update the manuscript for consistency.

Revised.

Lines 368-371: Not a comment to address, but I learned that levoglucosan can be present in coal combustion!

Thanks for pointing this out.

Line 396: How well does this linear model work? I'm not seeing anything demonstrating the quality of fit (e.g., $R^2$) or even a plot.

The $R^2$ is now added.

Figure 6: I suggest that the authors "jitter" either winter or summer to make the error bars clearer (e.g., shift one of them ½ hour to the left or right along with a note in the caption)

Figure 6 is now revised according to the comment.

Referee 2

Overall, I find this to be a largely observational paper that provides some interpretation. I find it to be written in such a way that it is often difficult to follow. Most often this is because of a too-rapid going back and forth between winter/summer. I think that the authors should strongly consider trying to organize each section to fully describe each season, and then make comparisons. This would also really help in instances when they are trying to make specific points about specific seasons.

I do also have some concerns about what the Esca-Dc plots mean when the Esca is < 1, and what this says about the uncertainty of the method overall. I agree with the first reviewer that the "what" is generally (although not always) clear, but the "why" is often lacking. I think the authors could do a better job at supporting their conclusions. There is a lot of very specific terminology used throughout and a summary table would be very helpful/welcome. I think that the measurements are, overall, of good quality and the analysis is likely robust, and thus the manuscript could ultimately be publishable. How-ever, I do also think that the manuscript could do with some structural reorganization within sections, with some greater details, and with stronger connections between observations and interpretation. My specific comments follow below, with ** put next to those that I think are more crucial.

We appreciate the very comprehensive comments from the referee. These comments have helped us greatly improve the manuscript. Here is a list of main modifications we have made according to the comments:

1) We have provided a detailed discussion below about $E_{sca}$ and what it represents when $E_{sca}$ is below 1.
2) The referee's detailed comments below focus mostly on re-balancing the shift of discussions between winter and summer is mainly for section 4.1. As we aim to compare the same properties for both seasons, we have discussed each of BC properties for both seasons then move to the next properties for a better context flow, rather than discussing all properties within one season.
3) We have revised our discussion relating to coagulation.
4) We have revised discussions on the comparison between modelled and measured absorption.

L34: The meaning of "dilution effect" is not clear.

We have revised as "The characteristics of the BC were relatively independent of air mass direction in summer; whereas in winter airmasses from the Northern Plateau were considerably cleaner and contained less-coated and smaller BC, whereas the BC from the Southern Plateau had the largest core size and coatings."

L36: The meaning of this sentence about source apportionment methods is not clear. What does "physical method" mean? And how are these combined if they are, apparently, performed separately? (Perhaps the text addresses this, but the abstract is unclear.)
We have revised this part accordingly (see response to referee 1).

L42: What does "tended to dominate with moderate coatings" mean? These particles do not dominate the BC mass overall. Are words missing?
"Dominate" is now replaced by "The BC from coal burning or biomass burning was characterized by…".

L100: The use of the word "novel" does not seem appropriate here. The techniques have been used previously. Perhaps a "novel combination" is appropriate, but even then I'm not certain as there are other studies that have looked at BC size distributions and composition.
The word "novel" is now removed.

References Liu et al. (2014a) and (2014b) are the same reference.
Corrected.

Fig. 1: The units on the emissions are not clear. It says Mg/m. Why per meter?
It is per month, revised.

L157: It is unclear how the authors established that this "is the optimum metric to reflect: : :" In what way specifically was it optimum? How was this established specifically? How is varying by +/-10% the right value?
The 10% and to vary +/-10% is an arbitrary manner, and even so the exact fraction to assign the air mass type based on air mass fraction does not have important impacts on the following analysis. We have added related information in the revised version.
"The 10% threshold was somewhat arbitrarily set based on an iteration over a range of values – too low a threshold biased air masses towards the very near field, whereas too high a value failed to classify a large fraction of air masses. The sensitivity of the threshold was tested by varying it between 9 and 11% (±10%) and the classification was shown to be insensitive to change."

L181: What is the smallest coating amount that can be reliably determined, given the uncertainty of the method?
The related information has been added.

"For the current SP2 configuration (Liu et al., 2017), the detection efficiency for the coating, taking into account only particles with sufficient signal to noise in the two element APD detector signal to perform reliable LEO fitting, is >80% for $D_c$=0.12-0.45μm, >60% for $D_c$ >0.45μm (due to partly saturated signal) and >50% for $D_c$=0.08-0.12 μm."

L195: I suggest the authors use a sub, rather than superscript for the MAC, such that there is zero ambiguity as to whether this is an exponent.
Changed.

L205: How was the PAX calibrated? How do the MAC values measured at 870 nm compare with those calculated at 550 nm?
We have added the related information in the revised version. We now compare the measurement and calculation both at 870nm. We have also added the following statement:
"The calibration of scattering and absorption for PAX was performed using the polystyrene latex spheres and fullerene soot respectively."

L223: What does it mean to apply PMF in "real time"? Per the cited Wang paper, the PMF analysis was conducted using standard methods, which are certainly not applied in "real time".
We have removed the phrase "real time".

L233: I am not certain that the statement "As the bottom panels show, the site was mostly influenced by northerly air masses in winter: : :" is justified by the data. The figure shows that probably half of the total period was dominated by Plateau South air masses.
We have revised the text to now read "the site was mostly influenced by northerly and westerly air masses in winter".

Fig. 3: The authors should consider using a different color scheme, especially one that does not put green and red next to each other.
We have changed the colour scheme.

L244: It would be helpful if the authors would elaborate on the meaning of the following sentence: "In summer, air masses from the western NCP showed lower RH which may result from the almost latitudinally homogenous distribution of higher temperatures." How does a "homogenous distribution" translate to lower RH? I understand why higher temperatures at the point of measurement might. But higher temperatures elsewhere could, at least in theory, lead to increased evaporation of water.
We removed the discussion.

L250: The authors state that rBC concentrations are higher in winter due to higher emissions. But boundary

layers are often also lower in winter, leading to higher concentrations of primary pollutants. How can the authors separate these effects, or at least rule out boundary layer differences as an important reason for the wintertime increase? This should especially be rationalized with the authors statement above that wintertime saw more air masses from the northern plateau yet that the northern plateau airmasses were linked to periods of the lowest concentrations. And since the authors note boundary layer height differences below (L269). They do this below, but they might either bring this discussion up or point the reader to the later discussion.

This text has now been revised.

"rBC mass loadings were higher in winter than in summer by around a factor of 2 for both local and regionally transported air masses, mostly likely due to a combination of higher surface emissions from both the local Beijing region and the surrounding area in the cold season, and also the increased frequency of lower boundary layer heights in winter (section 4.4)."

L256: When the authors state "This may be also: : :" the also makes me think that there has been some argument advanced already. But they do not seem to advance an argument before this point as to why the particles sizes from the northern plateau were smaller. Why would lower concentrations mean smaller sizes?

This sentence has been revised.

"This may result from more efficient removal processes for the more coated and larger BC particles in winter"

Fig. 5: If the authors were to show the averaged BC size distributions for the different air masses, perhaps as a supplemental figure, this would help the reader to understand the MMD histograms.

The figure has been added to the supplement as the referee suggested. (Fig. S2e.)

L264: It is not clear why "longer westerly transport" would result in larger rBC cores. This aspect needs to be justified.

The word "longer" has been removed.

** Section 4.1: I find that the authors bounce between summer/winter very quickly, and not always clearly. I suggest that this might be clearer if the authors were to fully present one season, and then the other, and then point out notable similarities/differences. As written, I find this more difficult to follow than it need be. This continues through many of the sections.

We tend to disagree with the referee on this point. The point of the section is to compare and contrast each BC property in the summer and winter seasons in order to link these to similarities or differences in sources or processes. In our view separating the two seasons makes it difficult to do this effectively so we have retained the original structure. We have, however, added two introductory sentences to the section to help the reader understand the structure more easily.

"The following section describes the measured BC properties in both the winter and the summer seasons.

Each property is discussed in turn and the similarities and differences between the seasons highlighted to clearly identify property changes that can be linked to changes in sources or processes in summer and winter."

L270: Just to be clear, when the authors refer to BC being "concentrated" by the shrinking of the PBL at night, they are not implying that the shrinking itself concentrates the BC, correct? This is not physically what happens. Emissions that do occur at night are into a smaller atmospheric region and thus end up more concentrated.

We thank the referee and changed the text to the following:

"In winter, surface cooling during the night leads to a shallow nocturnal boundary which, when coupled with increased emissions from heating activities greatly enhances surface pollution. During the day, a deeper boundary layer is re-established, resulting a much larger volume for emissions to mix into. The diurnal variation in rBC is a combination of both diurnal variations in emissions and marked changes in BL structure."

**BC Sources: Is BC from coal expected to be chemically similar as BC from other sources? I could see reasons it might be quite different chemically, and therefore quite different optically. If coal combustion, especially residential coal combustion, is an important source of wintertime BC, could it be possible that the coating amount estimation method might have some trouble, as it uses an RI that was determined for BC from (most likely) vehicle combustion? Is there any evidence available in the literature to illustrate that BC from coal behaves similarly and that the methods applied are appropriate? Has any direct source testing been done? I am unaware of any SP2 source sampling of coal. Yet, we know that the properties of BC can influence the SP2 measurements (e.g. Laborde et al.)

The referee makes a good point, there could well be chemical differences in BC from coal burning that lead to differences in optical properties compared to rBC from diesel exhaust emissions. It may also be that the burn conditions and the type of coal burned will also influence the rBC produced. To our knowledge there have been no direct measurements of rBC from coal combustion by the SP2. However, since the purpose of our analysis is not to predict the $E_{sca}$ quantitatively but rather to use the $E_{sca}$ vs $D_C$ distributions to identify differences in rBC properties from which we can identify different sources on a particle by particle basis. It therefore only matters that the distributions of sources are different from one another and this is indeed the case. We also note that the effects the referee has identified may also be the cause of some particles having Esca<1, since some sources may produce BC with a lower RI than traffic sources. We have added all of these discussions in the revised version (and also in response to further comments below).

"It is noted that $E_{sca}$ appears to be <1 at larger $D_c$. This arises for two reasons. Firstly, the assumption of sphericity used in the Mie calculations to derive $E_{sca}$ introduces significant bias for larger BC cores with little or no coating. The resulting scattering for uncoated BC will be lower than that derived by Mie-scattering due to geometrical influences as has been shown when comparing with T-matrix calculations (He et al., 2015;Wu et al., 2018) and this increases with rBC core size. Secondly, the BC at larger $D_c$ largely results from coal combustion (section 4.4), and this may have a refractive index (RI) that is different from

rBC from traffic sources (2.26,1,26). Both factors will lead to lower estimated scattering cross section for uncoated rBC, and hence $E_{sca}<1$. Since no measurements of BC morphology or source-dependent RI of rBC are available in this study, we are only able to state that these BC with larger $D_c$ is likely to be very thinly-coated and assume the particles with $E_{sca}<1$ had a $D_p/D_c=1$. This assumption is not likely to lead to major biases since the BC at large $D_c$ values has a lower MAC (Fig. S5) and adding material to these particles will only lead to the presence of a thin coating that will not significantly influence the resulting overall MAC (less than 8%)."

L274: I am not sure about the statement that there were no "obvious" diurnal variations in the BC coatings. The figure suggests a reasonably evident increase around 10 am in winter and a small increase around 9 am in summer. And in winter the size seems to be notably higher at night than day. The variation is not huge, but it seems evident. The authors use this statement regarding no "obvious" variations to argue that things are "well mixed during both seasons." This seems to be a bit of a stretch.
We have applied the statements suggested by the referee in the revised version.
"Fig. 6 shows an evident increase of BC coatings around 10 am in winter and a slight increase around 9 am in summer, and the coatings in winter were notably higher at night than in the day."

**L280: Is the conclusion that the size distributions are log normal robust over time? The two winter distributions shown suggest that if a campaign average were calculated one might need to use a multimode fit. Is this conclusion only true for a relatively narrow BC mass concentration range? That the MMD varies with BC concentration (Fig. 7b) suggests that the overall average distribution is not fittable by a single lognormal mode. Further clarification is needed. Also, are the fits given in Fig. 7b/c meaningful? These seem like arbitrarily chosen functions. Given the scatter in the data, one might think another functional form would work nearly as well. Are these just to show that the data vary? The authors might consider binning the data instead to illustrate this point.
We have clarified the Dc range used for the single lognormal fitting.
"as a single lognormal distribution fitted for $D_c$=100-400nm"
We have also tried to fit the distribution using two modes, however the second lognormal mode was subject to large uncertainty due to the saturation of the SP2 detector which has an upper cut off at Dc=550nm. This leads to insufficient data point constraints for the second moment fit. The following text has been added:
"However, fitting a second lognormal mode will be subject to large uncertainty due to the saturation of the SP2 detector which has an upper cut off at $D_c$=550nm, leading to insufficient data points to constrain such a fit."

An additional rBC mass distribution above 550nm may exist but this would require instrument reconfiguration to be fully detected. A two-moment lognormal fitting is therefore not performed in this study. We therefore only apply the single mode fitting in this study but clarify what has been done in the revised version.

We have removed the fitted function in Fig. 7b and c and replace them with an interpolated line between the binned average y-values.

**L295: The authors suggest coagulation might be responsible for the increase in BC size when the BC concentration is large. How can they exclude a shift in source? If they are going to speculate about one reason, they should speculate about the other reasonable interpretation (change in source). Also, arguments regarding coagulation would be strengthened if the authors could point to the fraction of the total particles that contain BC. Only BC-BC coagulation leads to growth. BC coagulation with non-BC particles does not lead to growth of the BC core. In many environments, BC is only a small fraction of the total particles. What is the situation here? I suggest this should be discussed. Especially, I don't understand how the authors can argue that a shift in sigma during one season is likely due to changes in source but in another it is coagulation just because the direction of the shift is different. This requires the authors knowing a priori what the end members of mixing line are with respect to width and MMD. The weakest part here, in my opinion, is in the range of concentrations where the two seasons overlap (0.1-4 ug/m3) and where different behavior is observed. Why would coagulation drive behavior during one season in the overlapping region but not the other? Especially when the authors argue for a "greater complexity of sources" in winter. I suggest this paragraph needs substantial revision.

We thank the reviewer for their valuable comments regarding our discussion of coagulation. We have revisited the dataset and have not observed a correlation between rBC mass loading and PM1 and the rBC mass fraction ranged between 1 and 10% regardless of the PM1 level (Fig. C1). We agree with the referee that if coagulation was important at high pollution levels, the BC could be coagulated with both BC and non-BC, the former will cause the increase of rBC core and the latter will result in the increase of BC coatings. The referee also points out that the observations may also reflect a shift in sources that is concentration dependent, whilst we do not observe any such concentration dependent sources in other analyses we have carried out this cannot be ruled out. A discussion of all of these possible contributions has now been added in the revised version.

The referee also points out the difficulties of ascertaining the different behaviour in summer and winter from the region of overlap. Winter has a larger $\sigma_g$ than in summer, which means that there was a wider range of $D_c$ values in winter than in summer. This may in part be attributable to the source influence; whereas at PM concentrations 0.5-4μg m$^{-3}$, the average $\sigma_g$ values were generally similar in both seasons. However, at higher pollution levels that were only observed in winter, the $\sigma_g$ decreased as a result of a narrower distribution. This may be partly explained by coagulation but may be also caused by a stronger source with larger Dc. These discussions are now added into the text.

"Fig. 7b shows the fitting parameters of BC core size distribution at different levels of rBC mass concentration. The core size generally increased at higher rBC mass concentration but demonstrated considerable variability ranging between 150-220nm. BC particles were observed to have systematically

larger core sizes in winter than in summer at the same rBC mass concentration. In winter, the significant increase of core MMD when the rBC mass concentration >5µg m$^{-3}$ may indicate that coagulation is taking place at high concentration. The increased BC coating thickness observed under these conditions is also consistent with BC coagulating with non-BC particles (Fig. 8). The width of the core size distribution $\sigma_g$ in winter showed a decreasing trend at higher rBC mass concentration, again this is consistent with the view that coagulation may occur at high rBC mass concentration reducing the width of the size distribution (Pratsinis, 1988). However, these effects may also reflect a shift in the range of sources present during periods of higher pollution levels that produces proportionally more rBC particles with large core sizes. While none of our other analyses indicate the presence of such a source, we cannot rule out this latter possibility and so are unable to unequivocally identify coagulation as the reason for these changes. In summer $\sigma_g$ showed an increasing trend with rBC mass concentration, which may result from more diverse source contributions at higher rBC mass concentration. The higher $\sigma_g$ in winter than in summer at the same rBC mass concentration suggests a greater complexity of sources in winter. The core sizes observed in Beijing are significantly larger ($p<0.01$) than those observed in London even when the BC source profile was dominated by wood burning (170nm), which may result from other sources of BC."

**L310: The discussion regarding the increase in coatings in the winter at higher BC concentrations would be, in my opinion, greatly strengthened if the authors also considered how the absolute concentrations of other PM species varied. Also, I find this discussion to be very weak in the context of the available data. The related paper (Wang et al.) uses PMF to analyze the coatings on BC. There are primary and secondary coating materials identified. How do these play into things? The discussion, as presented, is just statements of obvious factors that might impact coating amounts. But the authors could, and should, go beyond this, given the available data.

We thank the referee's suggestions on the discussion. Our discussion at this point in the paper serves to highlight the differences between different periods, Our analysis of the segregated rBC types that follows investigates these differences in more detail along the lines the referee suggests. To inform the reader of this we have included the following sentence:

"Possible reasons for these differences are investigated in more detail in sections 4.4 and 4.6 where the rBC is segregated into characteristic types and compared with PMF results from the SP-AMS and PM1 respectively."

**Fig. 9: How should one interpret the large number of points below the Esca = 1 line? Presumably, no point should be below this line if the interpretation is robust. This is especially important for the "large uncoated BC" region, which is almost entirely in a range where signal should not exist. Also, what is the smallest core size for which a coating can be reliably determined? (Is there any mismatch between the incandescence and scattering lower size detection limits?) I do understand that there is uncertainty in the measurements, and that perhaps this is what contributes. But the particular patterns of the Esca-Dc relationship, which trend towards values of Esca < 1 as Dc increases in general, really make me question the robustness of the method and interpretation.

The E$_{sca}$<1 at larger BC core size may result for the following reasons. 1) a larger BC core will introduce larger uncertainty when calculating the scattering properties since the Mie approach assumes the BC core is spherical. Previous work has shown that the scattering caused by uncoated BC will be lower than Mie-scattering using T-matrix calculations (He et al., 2015;Wu et al., 2018). Larger rBC will be subject to stronger geometrical influences so we would expect the discrepancy to increase with rBC core size. 2) the BC with larger D$_c$ results mainly from coal burning and these particles may have a RI lower than that from traffic sources (2.26,1,26). Both factors will lead to lower estimated scattering cross sections for uncoated rBC, and hence E$_{sca}$<1. As there is no constraint on the BC morphology or source-dependent RI of rBC in this study, we are only able to state that the rBC particles with low E$_{sca}$ values are likely to be very thinly coated and we assume the particles with E$_{sca}$<1 had a D$_p$/D$_c$=1. It is noted that the BC at such larger core size had a lower MAC (Fig. S5) and by adding slight coatings there will be no significant influence on the resulting MAC (less than 8%). These discussions are now included in the text and copied below. We also now state the lower detection limit of core size for the coating measurement in the revised version.

"It is noted that $E_{sca}$ appears to be <1 at larger $D_c$. This arises for two reasons. Firstly, the assumption of sphericity used in the Mie calculations to derive E$_{sca}$ introduces significant bias for larger BC cores with little or no coating. The resulting scattering for uncoated BC will be lower than that derived by Mie-scattering due to geometrical influences as has been shown when comparing with T-matrix calculations (He et al., 2015;Wu et al., 2018) and this increases with rBC core size. Secondly, the BC at larger $D_c$ largely results from coal combustion (section 4.4), and this may have a refractive index (RI) that is different from rBC from traffic sources (2.26,1,26). Both factors will lead to lower estimated scattering cross section for uncoated rBC, and hence $E_{sca}$<1. Since no measurements of BC morphology or source-dependent RI of rBC are available in this study, we are only able to state that these BC with larger $D_c$ is likely to be very thinly-coated and assume the particles with $E_{sca}$<1 had a $D_p$/$D_c$=1. This assumption is not likely to lead to major biases since the BC at large $D_c$ values has a lower MAC (Fig. S5) and adding material to these particles will only lead to the presence of a thin coating that will not significantly influence the resulting overall MAC (less than 8%)."

**Fig. S2: I do not understand this figure. The underlying distributions appear to have arbitrary sizes. I think these are somehow derived from the Esca-Dc relationship. But they are very oddly shaped, essentially unphysical. This figure is mentioned briefly as supporting conclusions regarding sources. But given that the shapes of the underlying distributions are so very, very strange, I believe that it requires substantial additional discussion. These are most clearly not log normal, as stated by the authors.
We have modified this plot by merging the size distribution of BC$_{sm}$ and BC$_{lg,uncoat}$, and now the three combined lognormal distributions are more meaningful and support our discussions.

L342: This should clarify that these are linear fits.
Corrected.

Section 4.4: It would be helpful to clarify that this is only for winter right at the start of this section.
This has been included in the first sentence of the section.

Section 4.4: The authors need to clearly define what they consider good, moderate, poor, etc. correlations. A value of R2 > 0.6 is stated as both "high" and "moderate" and "tightly" for example. Use of consistent language would facilitate consistent interpretation.
Corrected. The inconsistent descriptions have been removed values of $r^2>0.6$ have been to indicate the close correlation.

**L355: I am finding it difficult to understand all the terminology. The authors are variously defining things by ranges (I-IV), types (fossil fuel, biomass burning, traffic), etc. An effort to really clarify all the terminology would be most welcome and would facilitate the readers understanding. This is especially true when the authors make statements such as that the fossil fuel and biomass burning have similar core sizes and coating contents. I am having a very difficult time understanding what, specifically, the authors refer to (especially when I look again at Fig. S2, where the distributions of the different types seem to vary quite greatly).
We have now revised the terminologies throughout the texts to be $BC_{sm}$, $BC_{mod}$, $BC_{thick}$, $BC_{lg,uncoat}$. Fig. S2 has been modified to be made more clear.

OOA: Per the complementary Wang et al. paper, the OOA2/BC ratio is notably larger than the OOA1/BC ratio. Yet the OOA2_BC factor is more correlated with "moderately" coated BC while the OOA1_BC is more correlated with "thickly" coated BC. Can these be reconciled?
We don't understand what referee means "reconcile" here.

L374: This sentence could be rewritten to make it clearer. Use of commas, at least, would help.
Corrected.
"The $BC_{thick}$ or $BC_{mod}$ fraction is well correlated ($r^2>0.6$) with the BC coated with less-volatile organics (OOA1_BC) or semi-volatile organics (OOA2_BC), respectively."

L382: The wintertime BC was, generally, anticorrelated with the MLH. It did not "follow" the MLH.
Corrected.

L401: The citation to Xu et al. (2000) just points out that NOx can be controlled from coal combustion. It does not address the extent to which modern NOx controls are implemented today in and around Beijing. A more modern reference of direct relevance would be welcome.
Two recent references have been added per suggestion.
"The $BC_{lg,uncoat}$ was not correlated with $NO_x$ emission ($r^2<0.3$), which in turn suggests the coal combustion may not emit significant $NO_x$ (Zhao et al., 2013;Wang and Hao, 2012)".

L404: Since no large, uncoated BC mode was observed in London, and since this makes up a large fraction of the total, is it fair/relevant to compare the absolute fractions from this study to a study from London? I am not sure that it is. The authors might consider renormalizing, removing the large, uncoated BC mode from the statistics, if they wish to compare in this semi-quantitative way.

The referee refers to the comparison between the small, thinly coated rBC we make between Beijing and London. We feel that this comparison is reasonable since the small particles dominate the total number. We have included "number" to make this explicit when referring to the fraction.

Fig. 9: Is each panel individually normalized to the maximum?

Each panel has not been normalized but the colour scale is set to be red when the point density is above 70% of the maxima in each panel. This has now been clarified.

**Section 4.6: I am surprised to not see a discussion of how, perhaps, the small, uncoated BC is converted to thickly coated BC when the PM levels increase. It is evident from Fig. 12 that the sharp drop in the fraction of small, uncoated BC results largely from an increase in the thickly coated BC. What I find in this section is largely just a statement (or really, a series of statements) as to how the fractions of one type change with another. But there do not seem to be a lot of insights that actually come out of this section, in my opinion. I suggest that the authors focus more on development of insights rather than just a statement of relationships. Where they do try to develop insights, they really come off as speculative (e.g. L447) rather than fully developed.

We are not able to unequivocally determine the reasons for the relationships we observed and so did not speculate in the original paper. Nevertheless, we understand the referee's point and have added the following text:

"Given the complexity of the sources contributing to BC in Beijing, the relative primary source contributions and its interaction with other aerosol species may vary with the overall level of pollution and this in turn may change both the mixing state and the optical properties of BC. Fig. 12 shows mass fractions of different BC types at different $PM_1$ level determined by the total mass of AMS+SP2. The traffic-like $BC_{sm}$ (Fig. 12a) were a constant fraction of the total mass at lower pollution levels (when PM1<50 µg m$^{-3}$) that was around 30% and 40% in winter and summer respectively. The decreased mass fraction of $BC_{sm}$ at higher pollution levels occurs in both seasons but is particularly marked in winter, and matched by an equal and opposite increase in the mass fraction of $BC_{thick}$ under high pollution loadings. The $BC_{lg,uncoat}$ fraction (Fig. 12b) was similar in magnitude to the $BC_{sm}$ fraction in summer across all levels of pollution, which means the coal-burning like BC was almost as important as the traffic source. In winter, the contribution of $BC_{lg,uncoat}$ mass was slightly higher than the traffic-like $BC_{sm}$ mass fraction whereas in summer the $BC_{sm}$ mass was more significant. At the higher pollution levels in winter, some of the $BC_{lg,uncoat}$ may be also contributed by coagulation.

The increase in the mass fraction of $BC_{thick}$ at higher pollution levels (Fig. 12d) and the commensurate reduction in the mass fraction of $BC_{sm}$ (Fig. 12a) is very striking, especially when PM1>100µg m$^{-3}$ with

rBC mass loading > ~2 μg m$^{-3}$. Under these conditions, BC$_{thick}$ may be up to 50% of the total rBC mass. The coatings on these BC$_{thick}$ particles were largely contributed by secondary species according to the SP-AMS analysis (section 4.4). In summer, there were no periods of very high pollution in excess of 100 μg m$^{-3}$ and the BC$_{thick}$ mass fraction was less than 10%, though the trends in the fraction of BC$_{sm}$ and BC$_{thick}$ with pollution loading is the same in both seasons. This is entirely consistent with coagulation of particulate under very high concentrations. Under such conditions the very high numbers of BC$_{sm}$ may coagulate rapidly with the large numbers of on-BC accumulation mode pollution aerosol that is composed largely of secondary material, this process can happen rapidly at high number concentrations and lead to the small BC particles gaining very think coatings with no change in the core mass distribution (see also Fig. 9). High concentrations of secondary precursor in the gas phase will exacerbate this process."

We have adopted the comments and add the following discussions in the revised manuscript: "BC$_{sm}$ was converted to be thickly-coated BC at higher pollution level for both seasons. In wintertime at high pollution level with PM >100μg m$^{-3}$, the contribution of BC$_{lg}$ and B$_{mod}$ had not been significantly modified but BC$_{sm}$ had been efficiently converted to BC$_{thick}$ by a rate of about 10% per 100μg m$^{-3}$ increase of PM1 level, however below PM1 100μg m$^{-3}$ the fraction of BC$_{sm}$ almost maintained. This may imply a mixing process that the pollution level has to be above a certain level to allow the small BC core to be coagulated with other non-BC particle or condensed with more condensable materials, to form higher coatings on BC. This threshold of PM1 level in summer was at about 60μg m$^{-3}$, which is lower than in winter."

Fig. 12/L455: if the calculations of absorption really account for single particle coating state, then I have a difficult time understanding how there is a linear translation between the particle fractional contributions (left axis) and the absorption contribution (right axis). This implies that there is a single, characteristic value for each BC type that the fractional contribution can be multiplied by. But this wouldn't seem to go with the single-particle analysis. Is the single particle analysis not applied at each point in time to understand the variability within the different classes, but instead applied as a class average? I think it is the latter based on the discussion, but this could be clearer.
We thank the referee's clarification on the data analysis. Yes, it is a class average. We have clarified this point in the revised version and changed the figure caption.

L461: How was the MAC at 550 nm determined? The measurements were made at 870 nm. I find this unclear. Also, people typically think of BC as having an MAC at 550 nm around 7.5 m2/g (see e.g. Bond et al. (2006)). Yet, the translation between the MAC and Eabs in Fig. 13 implies a smaller MAC was used with the measurements, and this is confirmed on L458, although there is a seemingly contradictory value on L465 (which is perhaps just demonstrating the inadequacy of Mie theory for calculation of MAC values.) Unless what the authors are doing is actually showing the measured Eabs and a modeled MAC. It is not clear what the authors have done here.
We revised the current version and now report the modelled results at λ=870nm to directly compare with the measurements. The related discussions have now been added.

" in addition the mean PAX-measured $MAC_{870}$ was higher than the modelled results by 20% at this pollution level. This suggests complex processes at this moderate pollution level where a large variability of BC coating content was present, but a single mixing scenario based on Mie-calculation is not able to fully explain the measurements of absorption. These results suggest a general increase in absorbing capacity of BC at higher pollution level, implying that at higher pollution level, both primary emission and secondary processing may both contribute to the coatings and subsequent increased absorption."

**Section 4.6 – Absorption: The authors report their MAC values, but provide very little interpretation. Some interpretation would be welcome. They cite the Zhang (2018b) paper as some support of the reasonableness of their observations. But, Zhang et al. (2018b) find Eabs values at the same wavelength that never fall 1.7 while here the authors find at about the same conditions values of 1.2. (Personally, I think there are substantial problems with the Zhang et al. (2018b) paper, but nonetheless there is an inconsistency that challenges the simple citing of this as support.) The total PM/BC ratio appears to increase with the PM1 concentration. Could there be additional brown carbon leading to the increase? Or do the authors think that the increase results from the coatings? Is this what they are trying to imply (but not stating directly) when they compare the observations to the calculations (although as I note above the origin of the observations at 550 nm is not clear)? I think the authors should be more explicit. Note also that there is only a "shadowing" effect (L470) if the coating is absorbing.

We agree with the referee's suggestions that the brown carbon may contribute to part of the absorption enhancement at $\lambda=550nm$. We have performed the calculations at 870nm, the same wavelength as that at which the PAX measurements were made. Brown carbon does not absorb light at these wavelengths. Our results do still show an increase in absorption efficiency with pollution loading and hence imply an enhancement in the efficiency with which the BC core is absorbing light under high pollution loads.
We have removed Zhang et al., 2018 as per the referee's suggestion. We have also investigated the PM1/BC ratio as a function of PM1 shown in Fig. C1 (below) but found no apparent trend, which means the increase of PM1 did not necessarily imply a greater fraction of brown carbon.
We have added the following discussions in the revised manuscript:
"Fig. 13 shows both modelled and PAX-measured $MAC_{870}$ at different pollution levels, the results agree to within 15%. For both seasons, when PM1<50 µg m$^{-3}$, the measured absorption efficiency of BC showed little dependence on the concentration of particulate matter with $MAC_{870}$ ~4.9m$^2$g$^{-1}$ and 4.7 m$^2$g$^{-1}$ in winter and summer respectively. The absorption at $\lambda=870m$ is deemed to be unaffected by brown carbon (Sun et al., 2007). The $MAC_{870}$ for uncoated BC was calculated to be ~4.3 m$^2$g$^{-1}$ for both seasons so the enhancement of absorption efficiency ($E_{abs}$) is calculated as the modelled $MAC_{870}$ normalized by this value. $E_{abs,870}$ significantly increased at PM1>50 µg m$^{-3}$ up to 1.4 and 1.6 for summer and winter respectively. This increase in absorbing capacity of BC at higher pollution levels implies that both primary emission and secondary processing may both contribute to the coatings and subsequent increased absorption. It also highlights the potential for additional feedback between pollution and radiation under very high pollution loadings. There was a wide variability in MAC or $E_{abs}$ at PM1 concentrations of between 100 and 200 µg

m$^{-3}$. In addition, the mean PAX-measured MAC$_{870}$ was higher than the modelled results by 20% at this pollution level. This suggests complex processes at this moderate pollution level where a large variability of BC coating content was present, but a single mixing scenario based on Mie-calculation is not able to fully explain the measurements of absorption. The MAC$_{870}$ slightly decreased at very high PM1, i.e. >300 µg m$^{-3}$, and this decrease is more pronounced for measurement (the grey line shows) than Mie-based modelling. This may result from the shadowing effect that very thick coatings may shield incident photons from the absorbing core, particularly when the coating is absorbing (He et al., 2015;Zhang et al., 2018a).

"

[Figure]

Fig. C1. PM1/rBC mass ratio as a function of PM1 for both seasons.

References

He, C., Liou, K. N., Takano, Y., Zhang, R., Levy Zamora, M., Yang, P., Li, Q., and Leung, L. R.: Variation of the radiative properties during black carbon aging: theoretical and experimental intercomparison, Atmos. Chem. Phys., 15, 11967-11980, 10.5194/acp-15-11967-2015, 2015.

Wu, Y., Cheng, T., Liu, D., Allan, J. D., Zheng, L., and Chen, H.: Light Absorption Enhancement of Black Carbon Aerosol Constrained by Particle Morphology, Environ. Sci. Technol., 10.1021/acs.est.8b00636, 2018.

---

## Author Response (AR3)

Dear Editor and Referee,

We thank the referee once again for the comprehensive comments. We have addressed the comments as follows. Our replies are in blue.

The authors have done a generally good job responding to the reviewer comments. I do still find that the paper could be easier to follow, but as the other reviewer did not seem to have the same difficulty I do not wish to belabour my suggestion about organizational structure. I would ask the authors to address the below additional concerns.

L320: I guess I don't see the need to state "While none of our other analyses indicate the presence of such a source…". I actually don't think it's consistent with the rest of the paper. The authors discuss explicitly that there are likely increased emissions in the winter when concentrations are high. This undoubtedly results in a shift in the relative contributions from different source types (e.g. vehicle versus home heating versus energy production, etc.). This is evident in the supplementary figure S1. The residential is substantially larger in winter. There is absolutely a shift in sources between summer and winter. And within a season there is absolutely a dependence on which regions contribute with BC concentration (Fig. 5), and the relative balance of sources differs between regions. These issues are even discussed in the previous section of the paper. I suggest this is removed.

We have adopted the referee's suggestion and this part has been removed in the revised version.

Line 320.

Fig. S2: While I find the revised Fig. S2 improved, it is still evident that in many cases the distributions are not log normal, in contrast to the statements in the manuscript. There are not three "log normal modes". The sm+lg,uncoat distribution itself has at least two modes in nearly every case. Is there a need to argue for log normality? Or am I missing something?

We have reworded this part and removed the "lognormal".
line 367.

Previous Reviewer Comment: OOA: Per the complementary Wang et al. paper, the OOA2/BC ratio is notably larger than the OOA1/BC ratio. Yet the OOA2_BC factor is more correlated with "moderately" coated BC while the OOA1_BC is more correlated with "thickly" coated BC. Can these be reconciled?

Clarified comment: The authors indicate they are unsure what I am asking for with this comment. Basically, I am saying that if OOA1/BC is larger than OOA2/BC, then why does "thickly" coated BC correlate well with OOA2_BC while the "moderately" coated BC correlates with OOA1_BC. Wouldn't one expect the more "thickly" coated BC to correlate better with the factor where the coating/BC is larger?

We have more explicitly made this statement in the revised version.

"Note that the absolute values of rBC mass from SP-AMS or SP2 had not necessarily corresponded with each other for the correlated factors, e.g. the OOA2_BC is higher than OOA1_BC but vice versa for the SP2 factors. This may result from different responses and sensitivities for the measurement techniques of both instruments at different levels of BC mixing state."

Line 418-422.

L492: The comma after "material" should be a period.
corrected.

L494: How do high concentrations of precursors "exacerbate" coagulation? What matters is not the precursors, but the particles. Are the authors trying to say that there may be more particles when the precursor concentrations are high? If this is the case they should just say this. Also, implicit in this is the idea that this will lead to nucleation and formation of new particles. Without nucleation, the precursors will form new secondary material on particles, but not influence the particle number concentration, which is what matters for coagulation. They might also consider including data (if they have it) or a reference indicating that higher mass concentrations are also correlated with higher number concentrations in the region. It is not a given that they will (although I suspect this is the case).

We thank for the suggestions and the discussions have been added in the revised version.

"High concentrations of secondary precursor during higher pollution episode in the gas phase may lead to nucleation and new particle formation, or size growth on pre-existing particles. A further analysis shows high number concentration of particles also corresponded with higher mass concentrations, indicating that the new particle formation and primary emission coincided with particle size growth. These processes will in turn promote the high coatings on BC."

Line 497-502.

L504: I have some difficulty understanding the argument that the different BC types have differing contributions due to their varying absorption efficiency alone, which is what is implied as stated. I think the authors should also indicate that variations in absolute contributions of these types will also matter and, likely, dominate the behavior here.

The discussion has been revised:
"implying that different BC types have varying absorption efficiency but the contribution on absorption is determined by both mass loading and MAC".

Line 512-513.

L507: It should be noted that there is no strong experimental evidence (or at least none that I'm aware of) that the absorptivity of atmospheric BC falls off at large sizes. This is something that happens in Mie theory, to be sure, but it's not clear that this actually occurs, or if it does that it occurs to nearly the same extent as Mie theory predicts. (This has to do with atmospheric BC not actually being spherical.)

We agree with referee and the related discussion has been added.
"$BC_{lg,uncoat}$ had a lower MAC because of its larger core size which is consistent with Mie calculation."

Line 515.

L505: It should be stated explicitly here that these are calculations, and not measurements, of the MAC. Also, given the shift to 870 nm in Fig. 13 (the wavelength of the measurements), it is not clear why the authors have retained using 550 nm here. They should also state explicitly that they assume non-absorbing coatings. I realize these points are made in the "methods" section but they should be repeated here.

The MAC550 is used here in order to be compared with many other studies. This information is now added in both method section and the discussions here:

"The contribution of absorption coefficient is calculated based on single particle information shown in Fig. S4 (note that the coating is assumed to be non-absorbing here)."

Line 509-510.

L511: The authors should state these not as ~, but with the mean and a meaningful metric such as the standard deviation.

Corrected.

Line 519-522.

L514: An estimated uncertainty for the 1.4 and 1.6 should be provided. As noted previously, the calculated MAC values may be low, since it is known that Mie theory often underestimates MAC values relative to measurements. Consequently, the Eabs values here, determined as a ratio between a measurement and a calculation, have a decent chance of being biased high.

The uncertainty is now calculated based on the calculated uncertainty of uncoated BC, and this information is added in the revised version.

Line 523.

L524: It is not "particularly" when the coating is absorbing. It is only when the coating is absorbing.

Corrected.

Table A1 is a welcome addition, but missing many of the abbreviations used throughout.

This is double checked and added in the new table.

Page 19.

[revised manuscript text omitted]